# Tollip coordinates Parkin-dependent trafficking of mitochondrial-derived vesicles

Thomas A Ryan* , Elliott O Phillips, Charlotte L Collier , Alice JB Robinson, Daniel Routledge, Rebecca E Wood, Emelia A Assar & David A Tumbarello**

## Abstract

**Multiple mitochondrial quality control pathways exist to maintain the health of mitochondria and ensure cell homeostasis. Here, we investigate the role of the endosomal adaptor Tollip during the mitochondrial stress response and identify its interaction and colocalisation with the Parkinson's disease-associated E3 ubiquitin ligase Parkin. The interaction between Tollip and Parkin is dependent on the ubiquitin-binding CUE domain of Tollip, but independent of Tom1 and mitophagy. Interestingly, this interaction is independent of Parkin mitochondrial recruitment and ligase activity but requires an intact ubiquitin-like (UBL) domain. Importantly, Tollip regulates Parkin-dependent endosomal trafficking of a discrete subset of mitochondrial-derived vesicles (MDVs) to facilitate delivery to lysosomes. Retromer function and an interaction with Tom1 allow Tollip to facilitate late endosome/lysosome trafficking in response to mitochondrial stress. We find that upregulation of TOM20-positive MDVs upon mitochondrial stress requires Tollip interaction with ubiquitin, endosomal membranes and Tom1 to ensure their trafficking to the lysosomes. Thus, we conclude that Tollip, via an association with Parkin, is an essential coordinator to sort damaged mitochondrial-derived cargo to the lysosomes.**

**Keywords** lysosome; membrane trafficking; mitochondria; Parkinson's disease; vesicle transport
**Subject Categories** Membranes & Trafficking; Organelles
**The EMBO Journal (2020) 39: e102539**

## Introduction

Mitochondrial quality control pathways are essential to maintain their proper function and ensure overall cell health and survival. Dysfunction in these pathways is associated with several human pathologies, such as neurodegenerative disease and cardiomyopathy (Campos *et al*, 2016; Franco-Iborra *et al*, 2018). Mitophagy, the autophagic degradation of damaged mitochondria (Lemasters, 2005), is one of the most well-studied mechanisms of mitochondrial quality control and is an important pathway aimed at sequestering whole, damaged mitochondria to ensure their degradation in lysosomes. However, it is now understood that multiple routes of repair or whole degradation occur in response to mitochondrial insult (Pickles *et al*, 2018). The existence of these multiple mechanisms regulating mitostasis is likely critical in post-mitotic cells such as neurons where their extensive axonal arborisation requires variable bioenergetic demands (Misgeld & Schwarz, 2017), making them most susceptible to dysfunction in mitochondrial quality control.

A significant proportion of previous investigations have focused on PINK1/Parkin-mediated mitophagy. Upon mitochondrial damage, the serine/threonine kinase PINK1, which is normally rapidly imported into functional mitochondria from the cytosol and degraded (Jin *et al*, 2010), accumulates on the mitochondrial outer membrane (MOM) where it acts to recruit and activate Parkin (Matsuda *et al*, 2010; Narendra *et al*, 2010b), an E3 ubiquitin ligase, leading to the ubiquitylation of a number of MOM proteins (Gegg *et al*, 2010; Narendra *et al*, 2010a; Tanaka *et al*, 2010; Sarraf *et al*, 2013), thus ensuring its encapsulation and trafficking along the autophagic pathway. However, more recent data have highlighted that a number of alternative mitochondrial quality control pathways exist that require distinct membrane sources and are separate from canonical mitophagy (Xu *et al*, 2011; Soubannier *et al*, 2012; Abeliovich *et al*, 2013; McLelland *et al*, 2014, 2016; Lazarou *et al*, 2015; Hughes *et al*, 2016; Burman *et al*, 2017; Chakraborty *et al*, 2018; Di Rita *et al*, 2018; Rakovic *et al*, 2018). To date, it is not understood whether these distinct pathways function in parallel nor the extent of their conservation of mechanisms.

One such repair mechanism that delivers damaged portions or oxidised proteins to the lysosome is the mitochondrial-derived vesicle (MDV) pathway. These MDVs are formed following their budding from mitochondria and mediate the trafficking of specific mitochondrial cargo, such as oxidised proteins or lipids, to lysosomes (Soubannier *et al*, 2012; Wang *et al*, 2016) or peroxisomes (Neuspiel *et al*, 2008; Braschi *et al*, 2010) for degradation prior to fragmentation of the mitochondrial network (Cadete *et al*, 2016). Importantly, these events occur independent from mitophagy and fission machinery, but still depend on PINK1/Parkin function (McLelland *et al*, 2014). Although recent data have greatly increased

Biological Sciences, University of Southampton, Southampton, UK
*Corresponding author. Tel: +44 1517 954989; E-mail: t.ryan@liverpool.ac.uk
**Corresponding author. Tel: +44 2380 594288; E-mail: d.a.tumbarello@soton.ac.uk

our understanding of these pathways, it is still unclear how MDVs are formed, what defines their specificity for discrete mitochondrial cargo, and how they intersect with the endosomal pathway.

The precise mechanism of MDV trafficking has yet to be fully defined. However, it is clear that the retromer pathway mediates trafficking of at least some cargo-specific MDVs to their respective degradative compartments (Braschi *et al*, 2010; Soubannier *et al*, 2012; Wang *et al*, 2016). Mutations in the retromer subunit VPS35 have been shown to cause familial late-onset Parkinson's disease (PD) (Vilarino-Guell *et al*, 2011; Zimprich *et al*, 2011) and have also been detected in a sporadic case of the disease (Zimprich *et al*, 2011). Furthermore, Parkin directly interacts with VPS35 in *Drosophila* (Malik *et al*, 2015) and mammalian cells (Williams *et al*, 2018). The PD-associated D620N VPS35 mutation perturbs its interaction with WASH (Zavodszky *et al*, 2014), a protein complex involved in regulating endosomal protein sorting (Derivery *et al*, 2009; Gomez & Billadeau, 2009).

More recently, details of how the endosomal system interconnects with and regulates the individual mitochondrial quality control pathways have developed. Parkin has been shown to utilise a Rab5 early endosomal pathway to mediate mitochondrial sequestration and degradation (Hammerling *et al*, 2017a). In addition, PINK1/Parkin-dependent MDVs (McLelland *et al*, 2014) are shuttled to lysosomes via the endosomal system, independently of the autophagic protein ATG5 or LC3 (Soubannier *et al*, 2012), and a subset of these MDVs require Syntaxin-17 to form a soluble NSF attachment protein receptor (SNARE) complex with SNAP29 and VAMP7 in order to mediate their fusion with the lysosome (McLelland *et al*, 2016). Nevertheless, due to their cargo specificity and the use of discrete trafficking routes, it would seem logical that the MDV system is a collection of numerous distinct pathways. Although the mechanisms that allow for structurally distinct MDVs to differentially shuttle selective cargo to specific destinations have yet to be elucidated, it is clear that single-membrane and double-membrane MDVs traffic outer membrane and inner membrane/matrix proteins, respectively, by relying on distinct machinery (Sugiura *et al*, 2014).

Tollip (Toll-interacting protein) is an essential coordinator of the endosomal compartment and interacts with the ESCRT-0 protein Tom1 to regulate cargo trafficking (Katoh *et al*, 2004). Both Tollip and Tom1 have been separately implicated in autophagic dysfunction in Alzheimer's models (Makioka *et al*, 2016; Chen *et al*, 2017) and are known to cooperate in endocytic trafficking of ubiquitylated cargoes (Yamakami *et al*, 2003; Xiao *et al*, 2015). In addition, Tollip directly functions to clear Huntington's disease-associated polyQ aggregates via autophagy (Lu *et al*, 2014). Additionally, Tom1 interacts with the actin motor Myosin VI to mediate autophagosome–lysosome fusion (Tumbarello *et al*, 2012), with both of these proteins being identified as Parkin interactors in an unbiased screen (Sarraf *et al*, 2013). We demonstrate herein that Tollip localises to and facilitates the movement of a subset of MDVs through the endolysosomal system to ensure their turnover. Tollip also interacts with and recruits Parkin to a vesicular compartment and regulates MDV lysosomal docking. Importantly, abrogation of this pathway resulted in the arrest of cargo-specific MDV trafficking. Therefore, these results identify an essential mechanism that coordinates the trafficking of damaged mitochondrial components along the endolysosomal pathway.

# Results

## Tollip depletion leads to MDV accumulation

Current literature indicates that numerous distinct mitochondrial quality control pathways exist to allow discrete separation and trafficking of mitochondrial constituents to specific subcellular compartments, but with limited elucidation of the organisational mechanisms behind these divergent pathways. Interestingly, Tollip functions as an organiser of endosomal positioning and cargo trafficking to facilitate vesicle maturation (Jongsma *et al*, 2016), indicating it may act as a regulator of mitochondrial-derived vesicle trafficking, which requires endosomal routing to the lysosome. Therefore, our initial aim was to establish whether Tollip played a functional role in these discrete mitochondrial quality control pathways. We therefore performed siRNA knockdown of Tollip in SH-SY5Y neuroblastoma cells (Fig 1A) in order to analyse changes in the mitochondrial network. SH-SY5Y cells are a widely used model system for investigating mitochondrial quality control pathways due to their endogenous expression of Parkin, which we observed translocating to mitochondria following their damage (Fig EV1A). To allow for the identification of both the mitochondrial network and cargo-selective MDVs (Neuspiel *et al*, 2008; McLelland *et al*, 2014), mitochondria were co-labelled for TOM20 and the pyruvate dehydrogenase complex (PDH) E2/E3 bp subunits, proteins localised to the mitochondrial outer membrane (MOM) and mitochondrial matrix, respectively (Fig 1B). Strikingly, the knockdown of Tollip induced a significant increase in the number of TOM20$^{+ve}$/PDH$^{-ve}$ MDVs evident under steady-state conditions, which were maintained following mitochondrial stress (Fig 1C). This is supported by data in HEK293T Tollip KO cells generated by CRISPR-Cas9, which illustrates an accumulation of TOM20$^{+ve}$/PDH$^{-ve}$ MDVs (Fig EV1B). In contrast, the loss of Tollip expression had no effect on the number of PDH$^{+ve}$/TOM20$^{-ve}$ MDVs in SH-SY5Y cells (Fig 1D). In agreement with previous data (McLelland *et al*, 2014), an increase in PDH MDVs was observed in wild-type cells following both antimycin A/oligomycin in combination (AO) ($P < 0.001$) and 25 μM antimycin A alone (AA) ($P = 0.002$) (Fig 1D). However, unlike the observations of McLelland *et al*, we observed a small but significant increase in the reciprocal TOM20 MDVs following stress induction by both mechanisms in wild-type cells (untreated versus AO, $P = 0.0184$; untreated versus AA, $P = 0.0234$) (Fig 1C). Further supporting a role for Tollip in an MDV pathway, GFP-Tollip colocalised with TOM20$^{+ve}$/PDH$^{-ve}$ MDVs under stress conditions in both SH-SY5Y cells (Fig 1E) and HeLa cells expressing HA-Parkin (Fig EV1C). Additionally, we were able to better resolve TOM20 MDVs using super-resolution radial fluctuations (SRRFs) (Gustafsson *et al*, 2016), to illustrate the accumulation of these structures in Tollip siRNA-treated cells (Fig 1F) at resolutions of around 110 nm.

As Tollip acts to coordinate the trafficking of several proteins, we evaluated whether its absence altered Parkin recruitment to mitochondria and subsequent ubiquitylation of mitochondrial cargo. We performed these experiments on isolated mitochondria harvested from AO-treated parental and Tollip KO HEK293T cells overexpressing HA-Parkin. Indeed, depletion of Tollip did not block Parkin recruitment to mitochondria or their bulk ubiquitylation (Appendix Fig S1A), consistent with no alteration in PINK1 or Parkin translocation observed by immunofluorescence microscopy

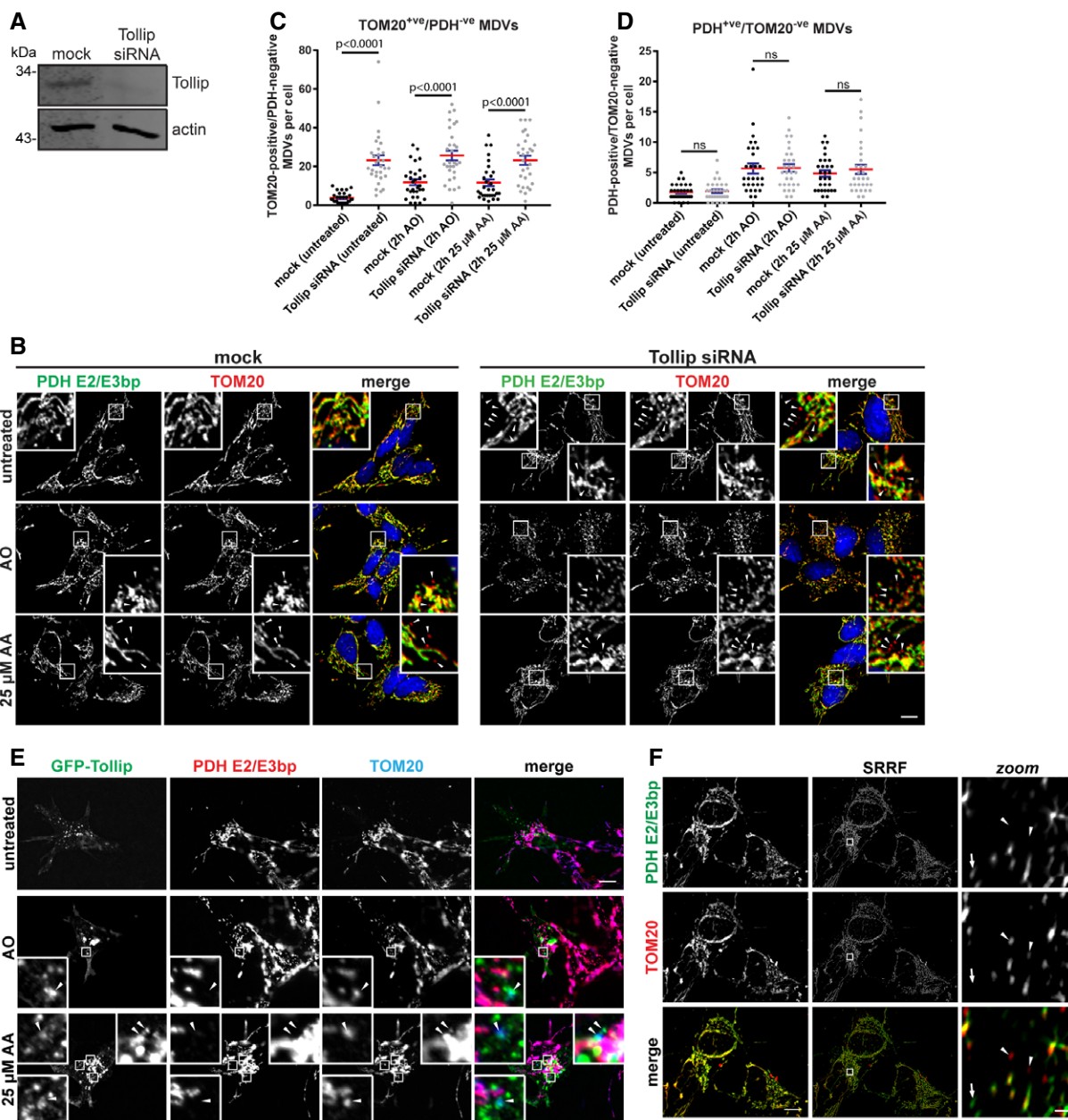

**Figure 1.  Tollip mediates the trafficking of TOM20+ve MDVs.**

A   siRNA knockdown of Tollip in SH-SY5Y cells was performed over 72 h, with cells reverse-transfected with siRNA oligos specific for Tollip (or mock) for 24 h before a second transfection was performed for a further 48 h. Knockdown was confirmed by Western blotting.

B   After knockdown of Tollip, SH-SY5Y cells were treated with 5 µM antimycin A/10 µM oligomycin (AO) or 25 µM antimycin A (AA) alone for 2 h, then fixed and stained for PDH E2/E3 bp (green), TOM20 (red) and nuclei (blue). Images were captured using widefield immunofluorescence microscopy. Zoom insets represent 4× magnification, and arrowheads indicate TOM20-positive MDVs.

C, D   The number of TOM20+ve/PDH−ve MDVs (C) or PDH+ve/TOM20−ve MDVs (D) per cell was quantified from immunofluorescence images captured (*n* = 30 cells per condition from two independent experiments). Statistical significance was determined using a two-way ANOVA (Tukey). Centre line indicates the mean, and error bars represent SEM.

E   SH-SY5Y cells were transfected with GFP-labelled Tollip for 24 h and then treated with AO, AO/100 µM bafilomycin A1 (AO/BfnA1) or AA alone for 2 h prior to fixation. Cells were then immunostained for GFP (green), PDH E2/E3 bp (red) and TOM20 (blue) and imaged by widefield immunofluorescence microscopy. GFP-Tollip was observed on TOM20+ve/PDH−ve MDVs (arrowheads). Zoom insets represent 7× magnification.

F   Images captured using super-resolution radial fluctuation (SRRF) microscopy showing both TOM20 (arrowheads) and PDH (arrows) MDVs in SH-SY5Y cells after Tollip siRNA knockdown. Left side images are the first in each of the original widefield image stacks. Scale bar for zoom images is 0.5 µm.

Data information: Scale bars are 10 µm, unless otherwise stated.
Source data are available online for this figure.

(Appendix Fig S1B–D). Together, these data suggest that loss of Tollip function leads to an accumulation of TOM20 MDVs, while having no effect on PINK1/Parkin translocation to damaged mitochondria.

### Tollip is in complex with and recruits Parkin onto a vesicular compartment

Parkin has been previously shown to regulate both mitophagy (Narendra *et al*, 2008) and selective MDV degradation pathways (McLelland *et al*, 2014). It would therefore be crucial to determine whether Tollip and Parkin functionally interact to regulate the MDV pathway. Firstly, we aimed to confirm whether Parkin is present on TOM20$^{+ve}$/PDH$^{-ve}$ MDVs. Overexpressed GFP-Parkin was previously shown not to localise to this class of MDVs, but rather to the reciprocal PDH-positive MDVs (McLelland *et al*, 2014). However, we observed the presence of endogenous Parkin on a subset of TOM20 MDVs in SH-SY5Y cells following Tollip depletion by siRNA (Fig 2A). To assess Tollip and Parkin colocalisation following induced mitochondrial damage, we transiently transfected SH-SY5Y cells with GFP-Tollip in order to observe its localisation with endogenous Parkin. Cells were subjected to mitochondrial stress induced by AO, with or without addition of the vATPase inhibitor bafilomycin A1 (BfnA1), in order to address the influence of endosomal trafficking on their vesicular association. Intriguingly, when we overexpressed Tollip we observed significant recruitment of Parkin onto a vesicular compartment illustrated by clear colocalisation of Parkin with Tollip (Fig 2B). A significant increase in this colocalisation occurred upon mitochondrial damage, which was perturbed when vATPase function was inhibited (Fig 2C and Appendix Fig S2A). Additionally, in SH-SY5Ys endogenous Tollip colocalised with endogenous Parkin in response to mitochondrial damage (Appendix Fig S2B), and this also occurred in HeLa cells coexpressing both GFP-Tollip and HA-Parkin (Appendix Fig S2C). Further

characterisation of this compartment indicated that a subset of these Tollip/Parkin vesicles are Rab7-positive (Fig 2D), verifying the endolysosomal nature of these vesicles and the requirement for endosomal maturation (Fig 2C).

To further elucidate the mechanism of this recruitment, we transfected cells with GFP-labelled Tollip R78A or M240A/F241A (CUE domain mutant) constructs (Appendix Fig S2D), which inhibit the binding of Tollip to phosphatidylinositol 3-phosphate (PI3P) and PI(4,5)P$_2$ (Torun *et al*, 2015) or ubiquitin (Appendix Fig S2E), respectively. Under steady-state conditions and following AO treatment, Parkin had an increased colocalisation with Tollip R78A compared with WT, whereas mutation of the CUE domain abolished this association (Fig 2E–G). As expected, the CUE mutant showed no colocalisation with ubiquitin inclusions, whereas R78A conversely demonstrated greater coincident staining with ubiquitin (Fig 2H). Interestingly, the level of Parkin/ubiquitin colocalisation differed between cells transfected with the individual mutants (Fig 2I), suggesting Tollip may function as a switch between its membrane and cargo binding in order to transfer damaged mitochondrial cargo along the endosomal pathway (Mitra *et al*, 2013). Indeed, R78A Tollip/Parkin vesicles show increased ubiquitin positivity compared with WT Tollip/Parkin vesicles (Fig 2J), indicating reduced membrane interactions may block progression of the pathway.

To test whether Tollip associates with Parkin in cells, we performed BioID proximity-dependent labelling in order to evaluate a Tollip–Parkin interaction. The BioID–Tollip fusion protein was stably expressed in HeLa cells alongside HA-Parkin. Cells were treated with AO to induce mitochondrial stress, and biotin labelling was performed for a 6-h period followed by streptavidin pulldowns to evaluate complex interacting partners. In addition to AO-induced mitochondrial stress, we also inhibited vATPase function (BfnA1) in order to block endosomal maturation. As expected, BioID–Tollip was able to biotinylate itself, as well as its known interaction partner

---

**Figure 2. Parkin and Tollip interact on a vesicular compartment.**

A   TOM20-positive MDV formation was induced in SH-SY5Y by siRNA knockdown of Tollip. MDVs were defined as TOM20$^{+ve}$/Cytochrome c (Cyt c)$^{-ve}$ via immunofluorescence microscopy, which indicated some colocalisation with Parkin (arrowheads).

B   SH-SY5Y cells were transfected with GFP-Tollip for 24 h and then treated with 5 μM antimycin A/10 μM oligomycin (AO) or AO/100 μM bafilomycin A1 (BfnA1) for 2 h prior to fixation and immunostaining. Antibodies specific to GFP (green), Parkin (red) and ubiquitin (blue) were used.

C   GFP-Tollip/Parkin colocalisation was quantified from cells under steady-state conditions or treated with AO and AO/BfnA1 by counting overall GFP puncta and GFP puncta positive for Parkin per cell, across 3–6 GFP-transfected cells/condition, per experiment (*n* = 3). Results are represented as a percentage of Tollip puncta positive for Parkin.

D   SH-SY5Y cells expressing GFP-Tollip were immunostained for GFP (green), Parkin (red) and Rab7a (blue). Arrowheads indicate areas of colocalisation between all three markers.

E, F   SH-SY5Y cells were transfected with GFP-Tollip R78A or CUE mutant for 24 h, and then treated with AO or AO/BfnA1 for 2 h prior to fixation and immunostaining, as described above for (B).

G   The percentage of GFP-Tollip puncta positive for Parkin was quantified from immunofluorescence images immunostained for GFP and Parkin from WT, R78A and CUE mutant expressing cells across 3–6 GFP-transfected cells/condition, per experiment (*n* = 3; *P = 0.0162, **P = 0.0034, ****P < 0.0001).

H   The percentage of ubiquitin puncta positive for Tollip was quantitated from immunofluorescence images immunostained for ubiquitin and GFP from WT, R78A and CUE mutant expressing cells across 3–6 GFP-transfected cells/condition, per experiment (*n* = 3; *P = 0.0382, **P = 0.0068).

I   The percentage of ubiquitin puncta positive for Parkin was quantitated from immunofluorescence images immunostained for ubiquitin and Parkin from WT, R78A and CUE mutant expressing cells across 3–6 GFP-transfected cells/condition, per experiment (*n* = 3; *P = 0.0232, **P = 0.0062).

J   The percentage of GFP-Tollip/Parkin vesicles positive for ubiquitin was quantitated from immunofluorescence images immunostained for GFP, Parkin and ubiquitin from WT, R78A and CUE mutant expressing cells across 3–6 GFP-transfected cells/condition, per experiment (*n* = 3; **P = 0.0070).

Data information: For analysis in (C) and (G–J), the number of GFP-positive puncta per cell ranged between 15 and 111 for WT Tollip, 14 and 56 for R78A Tollip and 0 and 45 for CUE mutant Tollip. Statistical significance was determined using a two-way ANOVA (Dunnett). Bar indicates the mean, and error bars represent SEM. Images were captured by widefield immunofluorescence microscopy at 100× magnification. Scale bar represents 10 μM. Zoom insets represent 3.25× magnification in (A) and 4.5× magnification in (B, D, E and F).

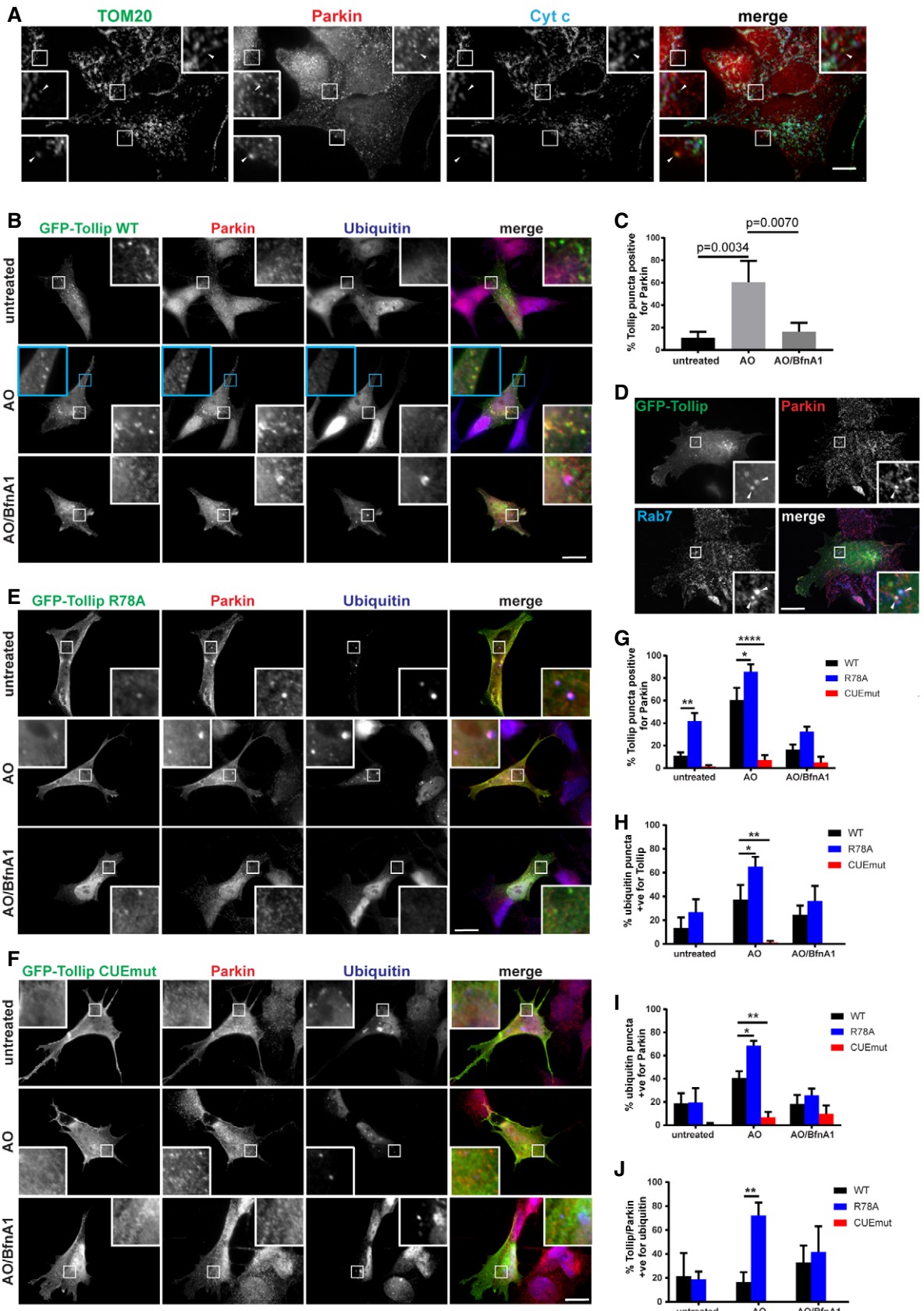

**Figure 2.**

Tom1, while BioID alone was only able to biotinylate itself (Fig 3A and B). Intriguingly, HA-Parkin is specifically biotinylated by BioID–Tollip under steady-state conditions, which is induced further following mitochondrial stress, while primarily unchanged in the presence of BfnA1 (Fig 3A). This was validated by coimmunoprecipitation,

which illustrates GFP-Tollip coprecipitating with HA-Parkin following AO-induced mitochondrial stress (Fig 3C), albeit weakly, suggesting a potential indirect or transient association. Although we observed trafficking of Tollip with MDVs (Fig 1E), no direct interaction with mitochondrial proteins is shown (Fig 3A).

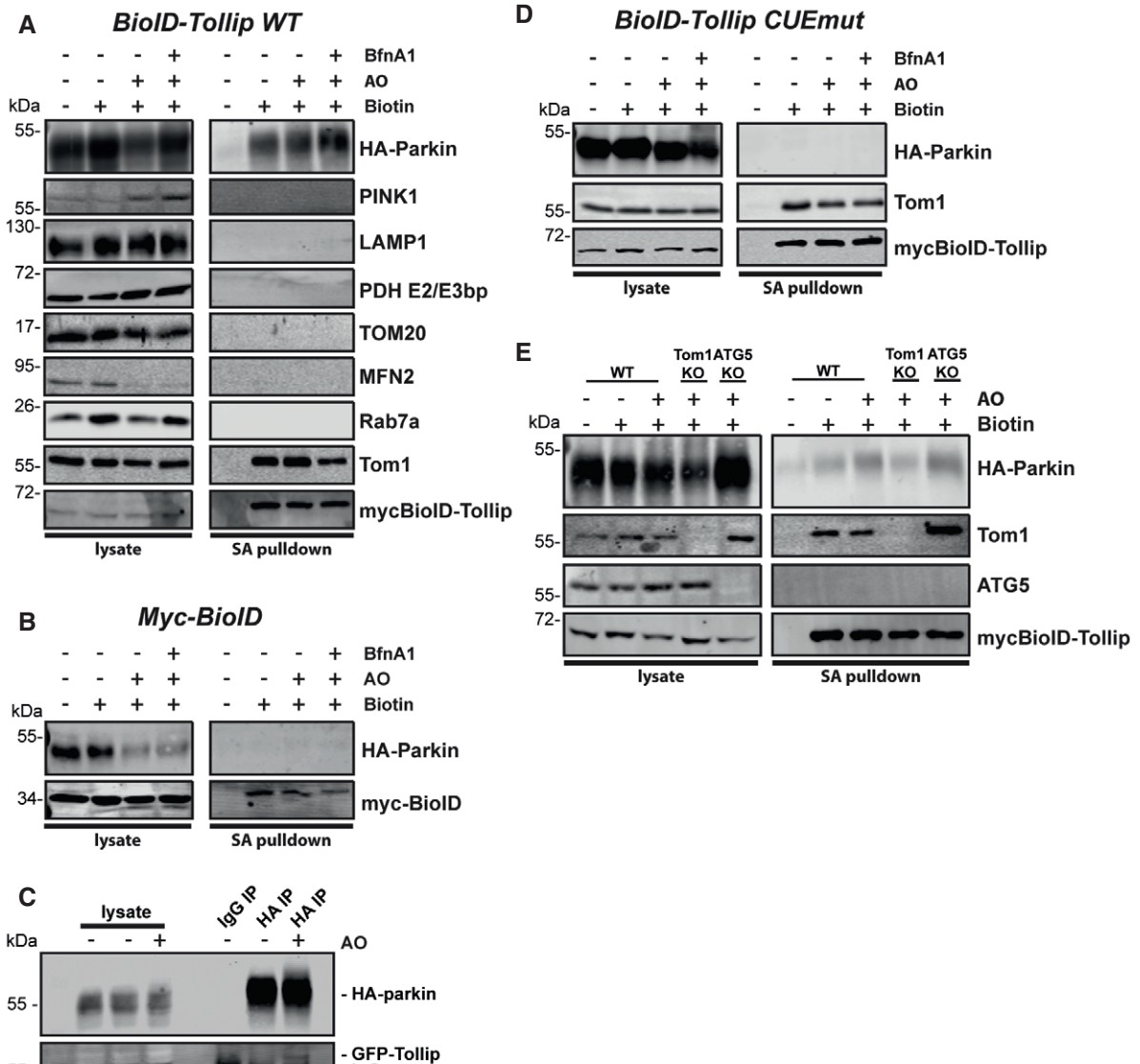

**Figure 3. Tollip and Parkin are in complex in a CUE domain-dependent manner, but independent of autophagy and Tom1.**

A  HeLa cells expressing mycBioID–Tollip and HA-Parkin were either left untreated or treated with 5 μM antimycin A/10 μM oligomycin (AO) or AO/100 μM bafilomycin A (BfnA1) for 6 h in the presence of biotin. Cells were lysed and streptavidin pulldowns performed overnight to isolate biotinylated proteins. Proteins in whole-cell extracts and pulldowns from each condition were then separated by SDS–PAGE and membranes probed with specific antibodies. Immunoblotting for Tom1 and mycBioID–Tollip was used as positive controls. Cells cultured in media lacking biotin were used as a negative control to assess background levels. Biotinylation of Parkin suggested Tollip specifically interacted with Parkin under these conditions.

B  HeLa cells expressing mycBioID alone and HA-Parkin were used to confirm the specificity of the Tollip–Parkin interaction.

C  HeLa cells coexpressing GFP-Tollip and HA-Parkin were either left untreated or treated with AO for 2 h prior to lysis. Cell lysates were subjected to either a normal rabbit IgG IP or HA IP, and Western blot analysis was performed using antibodies against Parkin and GFP.

D  HeLa cells expressing mycBioID–Tollip containing the CUE domain mutation (CUEmut) and HA-Parkin were subjected to biotinylation and streptavidin pulldowns, followed by Western blot analysis.

E  HeLa wild type (WT), Tom1 KO and ATG5 KO cells expressing mycBioID–Tollip and HA-Parkin underwent the same conditions as those above and immunoblotted for the indicated proteins.

Source data are available online for this figure.

To further analyse this interaction, we utilised cells expressing BioID conjugated to the ubiquitin-binding-deficient Tollip CUE mutant alongside HA-Parkin. Interestingly, this mutation abolished a Parkin–Tollip interaction (Fig 3D), suggesting ubiquitin binding may be required for the formation of this complex. Additionally, we observed that the Parkin–Tollip interaction is retained, close to wild-type levels, in both ATG5 KO and Tom1 KO cells (Fig 3E), confirming this complex is primarily formed independent of both autophagosome generation and a Tollip–Tom1 endosomal association, respectively.

## Parkin functions in TOM20 MDV trafficking

These data presented here suggest an interaction between Tollip and Parkin that regulates their spatiotemporal localisation in response to mitochondrial damage. Previous work has identified that Parkin is required for the formation of PDH$^{+ve}$/TOM20$^{-ve}$ MDVs, while not being present on TOM20$^{+ve}$/PDH$^{-ve}$ MDVs (McLelland *et al*, 2014). However, following Parkin siRNA-mediated knockdown we observed a significant increase in TOM20 MDVs under steady-state conditions, mimicking the phenotype induced by Tollip depletion (Fig 4A–C), while steady-state PDH MDVs were reduced (Fig 4D). This was corroborated using CRISPR-Cas9 generated Parkin KO SH-SY5Y cells (Fig 4E), which also indicated a specific elevation of TOM20 MDVs, while the level of PDH MDVs remains relatively unchanged compared with wild type (Fig EV2A–C). In addition, we did not observe a significant accumulation of TOM20 MDVs following siRNA knockdown of Tom1 (Fig 4C), indicating a limited requirement for Tom1, which in this case may be due to partial compensation by the related member Tom1L2 (Tumbarello *et al*, 2013). Importantly, these results together with data indicating an interaction between Parkin and Tollip on an endosomal compartment, as well as the presence of Parkin and Tollip on MDVs, would suggest that Parkin is not exclusively required for the formation of these TOM20 MDVs, but instead associated with their trafficking. In order to address this, we assessed the level of TOM20-positive MDV translocation to a LAMP1 compartment in Parkin KO cells, in order to determine whether their increased level was indeed due to a defect in trafficking to the lysosome. These data illustrate a substantial reduction in TOM20 MDV colocalisation with LAMP1 in response to AO-induced mitochondrial damage (Fig 4F and G), suggesting Parkin is required for TOM20 MDV trafficking to the lysosome. Furthermore, to exclude the possibility that loss of function of Parkin and Tollip results in increased mitochondrial damage triggering increased production of TOM20 MDVs, we evaluated the level of ATP production in SH-SY5Y Parkin and Tollip KO cells cultured in the presence of galactose. These results show similar levels of ATP production in both KO cell lines compared with wild-type cells, which is abolished upon oligomycin treatment (Fig EV3A and B), while exhibiting no variation in overall mitochondrial load illustrated by similar levels of mitochondrial resident proteins Mfn2 and TOM20 (Fig EV3C). Therefore, the wild-type levels of mitochondrial ATP production in the absence of Parkin and Tollip, in conjunction with data suggesting the cargo-selective nature of accumulating MDVs (Figs 1C and D, 4C and D, and EV2B and C), suggest excessive damage is likely not a significant contributing factor to TOM20 MDV accumulation.

To further analyse the connection between Parkin and Tollip, we expressed a number of PD-associated Parkin mutants in HeLa cells, which do not express Parkin endogenously due to the position of the *PARKIN* gene on a fragile site within chromosome 6 (Denison *et al*, 2003). We then analysed the localisation of these Parkin mutants by immunofluorescence microscopy, which indicated all Parkin constructs remaining cytosolic under steady-state conditions (Fig EV4A). However, as previously described (Matsuda *et al*, 2010; Narendra *et al*, 2010b), WT Parkin translocated to mitochondria upon uncoupling. Additionally, mutation of the ubiquitin-like (UBL) domain (R42P) still allowed for this recruitment, with previous

---

**Figure 4.  Parkin facilitates MDV trafficking and requires an intact UBL domain for Tollip association.**

A    Cells were reverse-transfected with Parkin or Tom1 siRNA oligos for 24 h before a second transfection was performed for a further 48 h, then fixed and stained with antibodies specific to TOM20 (green) and PDH E2/E3 bp (red). Nuclei are labelled with Hoechst (blue). Zoom insets represent 5× magnification, and arrowheads indicate TOM20-positive MDVs. Scale bar represents 10 μm.

B    Knockdowns were confirmed for both Parkin and Tom1 by Western blotting.

C, D    Quantification of MDVs, defined as TOM20$^{+ve}$/PDH$^{-ve}$ (C) or PDH$^{+ve}$/TOM20$^{-ve}$ (D) per cell under steady-state conditions, was performed (*n* = 32–52 cells per condition from three independent experiments). Statistical significance was determined using a two-way ANOVA (Dunnett) (ns = not significant). Centre line indicates mean, and error bars represent SEM.

E    Western blot analysis of lysates harvested from SH-SY5Y wild type and CRISPR generated Parkin KO cells. Immunoblot analysis was performed using antibodies against Parkin and GAPDH.

F    Immunofluorescence microscopy was performed on SH-SY5Y wild type and Parkin KO cells either untreated or treated with AO for 2 h. Cells were immunostained for TOM20 (blue), PDH E2/E3 bp (red) and LAMP1 (green). Zoom insets represent 5× magnification. Circles indicate TOM20 MDVs positive for LAMP1, and arrowheads indicate TOM20 MDVs negative for LAMP1. Scale bar represents 10 μm.

G    Quantitation of TOM20-positive MDV colocalisation with LAMP1 was performed across all treatment groups and represented as the % of TOM20-positive MDVs colocalising with LAMP1. Results represent three independent experiments (28–53 cells/condition), and statistical significance was determined using a two-way ANOVA (Sidak). Centre line indicates mean, and error bars represent SEM.

H    HeLa cells transiently transfected with HA-Parkin were subjected to mitochondrial stress by treatment with 25 μM antimycin A (AA) for 2 h, then fixed and stained with antibodies specific to HA, PDH and TOM20. TOM20$^{+ve}$/PDH$^{-ve}$ MDVs were quantified in cells that were either expressing or not expressing HA-Parkin (*n* = 12–14 cells per condition from three independent experiments). Statistical significance was determined using a two-way ANOVA (Sidak). Centre line indicates the mean, and error bars represent SEM.

I    HeLa cells stably coexpressing myc BioID–Tollip and either HA-Parkin WT or Δ1–76 (ΔUBL) were incubated with and without biotin in the presence of AO for 6 h prior to being subjected to cell lysis and streptavidin pulldowns. Western blot analysis was performed on lysate inputs and streptavidin pulldowns (Streptavidin PD) using antibodies against the indicated proteins or epitope tags.

Source data are available online for this figure.

---

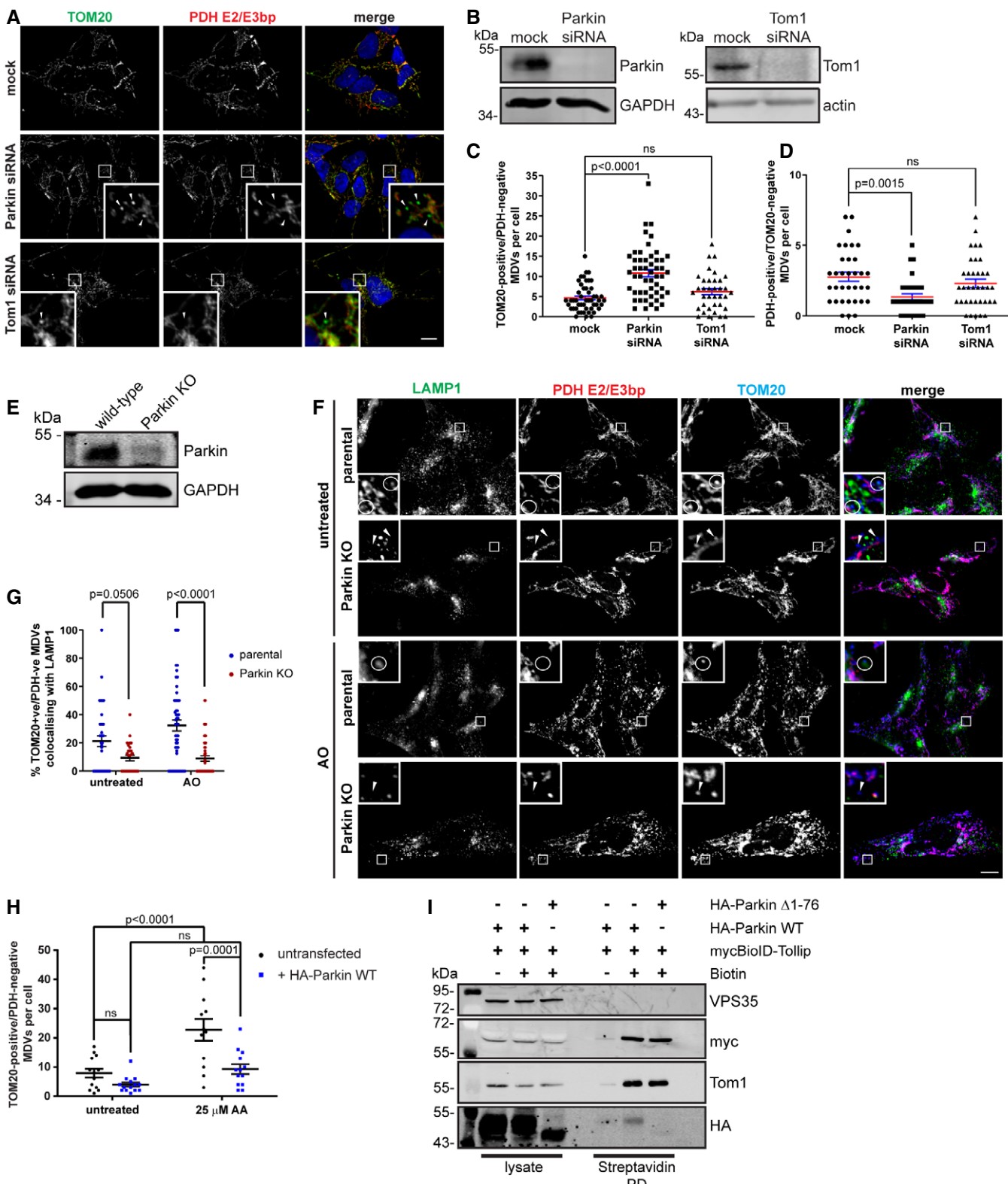

Figure 4.

studies demonstrating this mutation maintains E3 activity, albeit at a lower level, but disrupts its autoinhibited state (Chaugule *et al*, 2011; Yi *et al*, 2019). Again in agreement with prior studies,

mutations T240R and C431S abolished mitochondrial translocation (Geisler *et al*, 2010; Lazarou *et al*, 2013), which also render them E3-deficient unlike R275W which, despite still translocating here

(Fig EV4A), has markedly reduced E3 ligase capacity (Koyano *et al*, 2013).

To further elucidate the mechanism of the Tollip–Parkin interaction observed here (Fig 3A), we created HeLa cell lines stably expressing both BioID–Tollip alongside these Parkin mutants to determine whether specific point mutations can disrupt this complex. Interestingly, mutations abolishing Parkin E3 ligase activity or mitochondrial recruitment did not perturb interaction with Tollip (Fig EV4B). Moreover, unlike PDH MDV generation, which requires Parkin E3 ligase activity (McLelland *et al*, 2014), the reciprocal TOM20 MDVs investigated here (TOM20$^{+ve}$/PDH$^{−ve}$) were still observed in HeLa cells expressing E3-deficient Parkin mutants (Appendix Fig S3A), corroborating our previous Parkin siRNA data (Fig 4A–C), indicating Parkin is not required for the formation of this specific MDV subtype. Instead, our data collectively indicate Parkin is essential for TOM20 MDV trafficking as illustrated by experiments which show both Parkin knockdown and genetic knockout in SH-SY5Y cells lead to an accumulation of MDVs (Figs 4C and EV2), while in Parkin-deficient HeLa cells MDV load is elevated following AA treatment, which is subsequently reduced following re-introduction of Parkin (Fig 4H and Appendix Fig S3B). Furthermore, the R42P mutant, which disrupts UBL domain folding and the autoinhibited state (Safadi & Shaw, 2007; Chaugule *et al*, 2011; Liu *et al*, 2015), still maintained an interaction with Tollip (Fig EV4B); however, deletion of the Parkin UBL domain (Δaa1–76) resulted in a significant reduction in a Tollip interaction (Fig 4I). This relationship with the UBL domain is similar to previous reports characterising the interactions of Parkin with the endosomal adaptor Eps15 (Fallon *et al*, 2006). Therefore, these data suggest the association of Tollip with Parkin does not require Parkin's ligase activity or mitochondrial recruitment, but instead requires an intact UBL domain.

## Tollip is essential for TOM20 MDVs to traffic through the endosomal system

Although Tollip is not in direct complex with Rab7 (Fig 3A), it was apparent that a proportion of these Tollip/Parkin vesicles trafficked through a Rab7-positive compartment (Fig 2D). This was particularly interesting as Rab7-mediated trafficking has recently been

linked to mitochondrial maintenance through coordination of the mitophagy machinery (Jimenez-Orgaz *et al*, 2018) and regulation of mitochondria–lysosome contact sites (Wong *et al*, 2018), as well as its activity being modulated by Parkin (Song *et al*, 2016). Furthermore, Rab7a and Rab5 are recruited to damaged mitochondria during Parkin-mediated mitophagy (Yamano *et al*, 2018), with the latter being utilised to mediate an endosomal mitochondrial clearance pathway distinct from autophagy (Hammerling *et al*, 2017a,b). Importantly, Tollip can function as a modulator of protein sorting through its interaction with ubiquitin (Yamakami *et al*, 2003) and has been shown to localise to both early (Katoh *et al*, 2004) and late endosomes (Brissoni *et al*, 2006), to organise endosomal positioning (Jongsma *et al*, 2016).

To determine where TOM20$^{+ve}$/PDH$^{−ve}$ MDVs entered the endosomal system, we utilised a GFP-2xFYVE construct to investigate these early-stage events in cells. The FYVE domain is a double zinc finger that binds phosphatidylinositol 3-phosphate (PI(3)P) (Burd & Emr, 1998), which is enriched on endosomal membranes, with high specificity. To induce MDV formation, SH-SY5Y cells were treated with 25 μM AA for 2 h following siRNA knockdown of Tollip or Parkin, to assess whether the MDVs that accumulate under these conditions (Figs 1 and 4) acquire PI(3)P membranes. In order to observe the localisation of FYVE-positive membranes relative to MDVs, cells were transfected with GFP-2xFYVE prior to treatment. In wild-type cells, we observed a significant number of TOM20$^{+ve}$/PDH$^{−ve}$ MDVs that colocalised with the GFP-2xFYVE probe (Fig 5A–C) and the early endosomal marker EEA1 (Fig 5D), indicating these MDVs traffic through the endosomal pathway. Interestingly, loss of Tollip expression reduced MDV colocalisation with the GFP-FYVE probe (Fig 5C), indicating a block in their transition into the endosomal system. Conversely, depletion of Parkin expression did not disrupt the uptake of MDVs into the endosomal compartment and Tollip recruitment to MDVs was still evident (Fig 5C and E).

In order to understand Tollip's engagement with the MDV pathway, we investigated the endosomal trafficking route of Tollip in response to mitochondrial damage. SH-SY5Y cells were transfected with GFP-tagged Rab5 (early endosomes) or Rab7 (late endosomes) and subjected to mitochondrial stress. Under steady-state conditions, we observed a portion of the endogenous Tollip pool localised to Rab5 vesicles (Appendix Fig S4A), which was increased following

**Figure 5. Tollip facilitates endosomal "capture" of MDVs.**

A, B    siRNA knockdown of Tollip or Parkin in SH-SY5Y cells was performed over a 72-h period, and then 18 h prior to treatment, cells were transfected with GFP-2xFYVE to label endosomal membranes. Cells were treated with 25 μM antimycin A (AA) for 2 h, then fixed and immunostained with antibodies specific to PDH E2/E3 bp (red) and TOM20 (blue). Arrowheads denote TOM20$^{+ve}$/PDH$^{−ve}$ MDVs that colocalise with GFP-2xFYVE, and arrows denote TOM20$^{+ve}$/PDH$^{−ve}$ MDVs that do not. Zoom insets represent 4.5× magnification.

C    Quantification of TOM20$^{+ve}$/PDH$^{−ve}$ MDVs that are positive for GFP-2xFYVE is shown as a % of TOM20 MDVs positive for GFP, with each data point indicating 1 cell. MDV colocalisation was quantified by eye from images captured (*n* = 20–22 cells per condition from three independent experiments). Statistical significance was determined using a two-way ANOVA (Dunnett). Centre line indicates the mean, and error bars represent SEM.

D    SH-SY5Y cells were left untreated or treated with 25 μM AA for 2 h prior to fixation and immunostaining for EEA1 (green), PDH E2/E3 bp (red) and TOM20 (blue). Arrowheads indicate TOM20$^{+ve}$/PDH$^{−ve}$ MDVs positive for the early endosome marker EEA1. Zoom insets represent 5× magnification.

E    siRNA knockdown of Parkin was performed in SH-SY5Y cells followed by 25 μM AA for 2 h prior to fixation and immunostaining with antibodies specific to TOM20 (green), Tollip (red) and Cytochrome c (blue). We observed Tollip colocalisation with TOM20$^{+ve}$/PDH$^{−ve}$ MDVs (denoted by arrowheads), which was still maintained following Parkin siRNA knockdown. Zoom insets represent 5× magnification.

F    SH-SY5Y cells were transfected with GFP-Tollip and treated with 5 μM antimycin A/10 μM oligomycin (AO) or AO/100 μM bafilomycin A (BfnA1) for 2 h prior to fixation and immunostaining with antibodies specific to GFP (green), Rab7a (red) and PDH E2/E3 bp (blue). Zoom insets represent 4× magnification.

G    Quantification of GFP-Tollip/Rab7a colocalisation using Pearson's correlation. Each data point represents 1 cell (12–14 cells/condition from three independent experiments), and bar indicates the mean. Statistical significance was assessed using a one-way ANOVA (Dunnett). Error bars represent SEM.

Data information: Scale bars represent 10 μm.

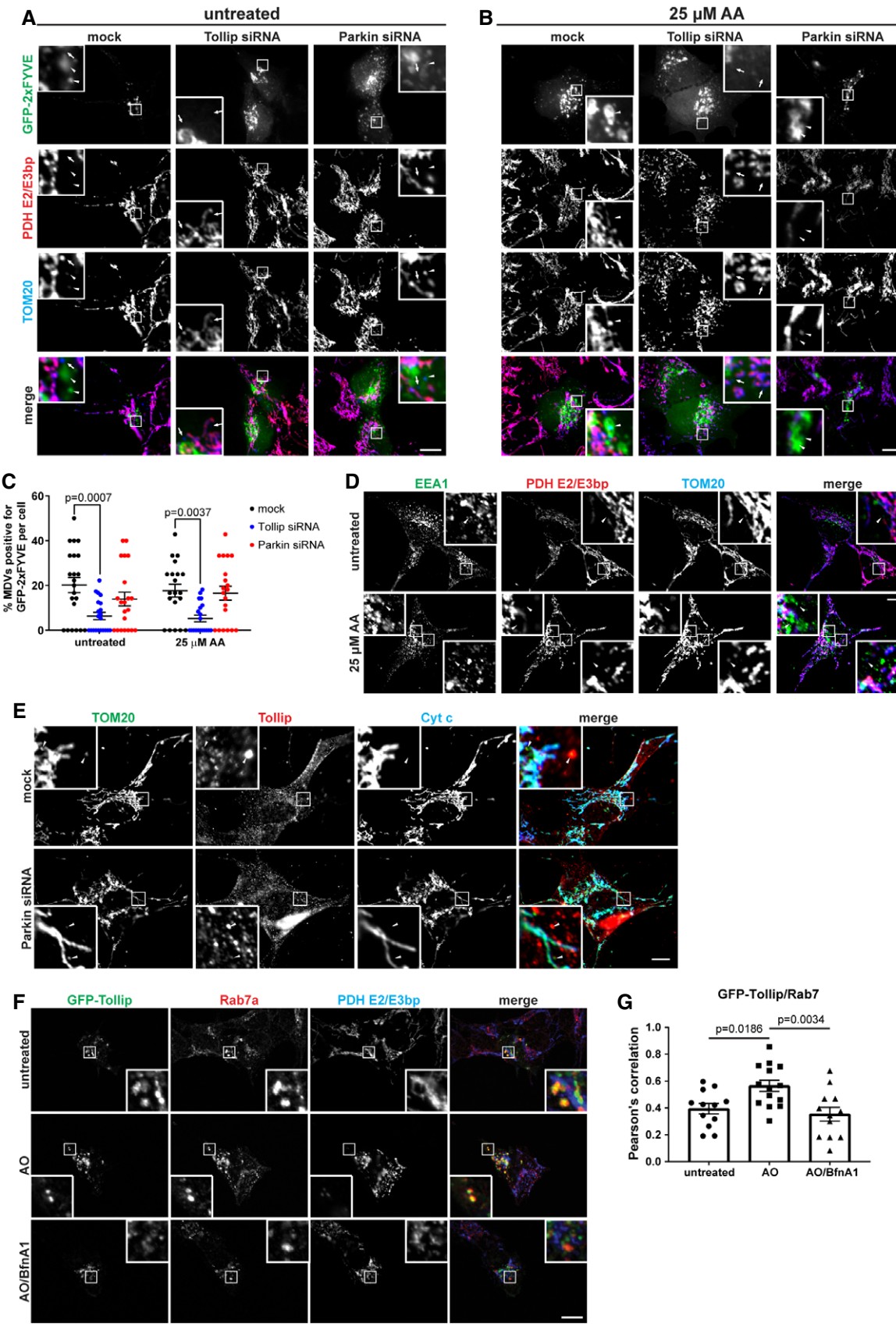

Figure 5.

mitochondrial damage in the presence of bafilomycin A1, but not apparent in cells treated with AO alone. In contrast, a subpopulation of Tollip was localised to Rab7-positive late endosomes following AO alone, but not apparent in cells treated with AO in combination with bafilomycin A1 (Appendix Fig S4B). We also validated the Tollip translocation to a late endosomal compartment, by evaluating its colocalisation with the endogenous pool of Rab7. Indeed, we observed an increase in the colocalisation of Tollip and endogenous Rab7a after treatment with AO, which was blocked by bafilomycin A1 and distinct from the mitochondrial network (Fig 5F and G). In addition, a subset of these MDVs produced following AO insult were positive for GFP-Rab7 (Appendix Fig S4C), further demonstrating their trafficking along an endosomal route. These data would suggest that localisation of Tollip to early and late endosomes is dependent on endosomal maturation and actively occurs upon mito-chondrial damage. Moreover, Tollip is essential for the integration of MDVs into an endosomal compartment.

### Tollip localisation to a LAMP1 compartment in response to mitochondrial stress requires interactions with Tom1, ubiquitylated cargo and membrane

Data presented here would suggest that Tollip and MDVs translocate through endosomal compartments during mitochondrial stress. As TOM20 MDVs are known to traffic to the lysosome (Soubannier *et al*, 2012), we aimed to assess Tollip localisation in relation to LAMP1. During AO-induced mitochondrial stress, we observed a clustering of endogenous Tollip in SH-SY5Y cells (Fig 6A), which was not associated with mitochondria (Appendix Fig S4D) or the Golgi (Appendix Fig S4E). Moreover, this induced clustering was replicated in HEK293 cells, but substantially suppressed in *TOM1*-deficient cells (Fig 6B and Appendix Fig S4F). We then determined these Tollip clusters were LAMP1-positive (Fig 6C), in line with previously reported data (Katoh *et al*, 2004, 2006; Baker *et al*, 2015). To further elucidate the trafficking of Tollip during the mito-chondrial stress response, we transfected SH-SY5Y and HeLa cells with various GFP-tagged Tollip constructs. Upon mitochondrial

damage, we observed an increase in the trafficking of Tollip to LAMP1 compartments (Fig 6D–F). Interestingly, deletion of the first 53 amino acids of Tollip (ΔNterm), which abolishes its binding to Tom1 (Appendix Fig S5A), or mutation of its CUE domain to block ubiquitin binding (Appendix Fig S2E), significantly suppressed its accumulation on LAMP1 compartments (Fig 6D and E). This would further implicate Tom1, and additionally ubiquitylated cargo, as important mediators of Tollip lysosomal trafficking.

Importantly, this defect in Tollip trafficking was replicated in HeLa cells expressing the Tom1 and ubiquitin-binding mutants (Appendix Fig S5B–D) and in HeLa Tom1 KO cells (Fig 3E) express-ing wild-type Tollip (Fig 6G and Appendix Fig S5E) in agreement with data from HEK293 cells (Fig 6B). Lysosomal translocation of Tollip was, however, maintained in ATG5 knockout cells (Fig 6H and Appendix Fig S5F), indicating that Tollip delivery to a LAMP1 compartment does not occur via autophagosomes. Additionally, upon mitochondrial damage, Tollip translocated to more catalyti-cally active compartments, as indicated by its increased colocalisa-tion with cathepsin D (Appendix Fig S5G), correlating with increased LAMP1 colocalisation with cathepsin D (Appendix Fig S5H). These data are indicative of directed endosomal trafficking of Tollip during the mitochondrial stress response that is independent of canonical autophagy.

Similar to results observed here from Tollip siRNA knockdown, depletion of the retromer subunit VPS35 has previously been shown to result in the accumulation of TOM20 MDVs (Braschi *et al*, 2010). We therefore aimed to investigate whether Tollip and VPS35 func-tion in the same mitochondrial stress response pathway. Indeed, we observed a consistent level of colocalisation between GFP-Tollip and VPS35 in SH-SY5Y cells under both steady-state and stress condi-tions, which was disrupted by bafilomycin A1 treatment (Fig EV5A and B). However, we did not observe any recruitment of VPS35 to TOM20 MDVs induced by either mitochondrial damage or Tollip knockdown, although VPS35-positive vesicles were consistently observed adjacent to mitochondria (Fig EV5C). To dissect any regu-latory role for VPS35 in Tollip function, we created a VPS35 CRISPR-Cas9 knockout HeLa line. Interestingly, knockout of VPS35 did not

---

**Figure 6. Tollip translocates to a LAMP1 compartment during mitochondrial stress.** ▶

A   Quantitation of Tollip vesicular clusters in SH-SY5Ys treated with 5 μM antimycin A/10 μM oligomycin (AO) for 2 h. The percentage of cells with vesicular clusters was quantified from 7 to 23 cells for each condition from four independent experiments. Statistical significance was assessed using an unpaired *t*-test. Bars indicate the mean value, and error bars represent SEM.

B   Quantitation of the prevalence of Tollip clusters in HEK293 WT or Tom1 KO cells treated with AO for 2 h. The percentage of cells with vesicular clusters was quantified from an average of 50 cells for each condition from five independent experiments. Statistical significance was determined using a two-way ANOVA (Sidak). Bars indicate the mean value, and error bars represent SEM.

C   Endogenous Tollip and LAMP1 colocalise in SH-SY5Y cells treated with AO for 2 h. Antibodies specific to LAMP1 (green) and Tollip (red) were used. Zoom insets represent 2.5× magnification.

D, E   SH-SY5Y cells were transfected with either wild-type GFP-Tollip (WT), GFP-Tollip lacking its N-terminus (ΔNterm) or GFP-Tollip containing a CUE domain mutation (CUEmut) and then treated with AO for 2 h. Cells were fixed and immunostained with antibodies specific to GFP (green) and LAMP1 (red). Zoom insets represent 2.5× magnification.

F   The extent of colocalisation between GFP-Tollip and LAMP1 from untreated and AO-treated SH-SY5Y cells was quantitated from immunofluorescence images and represented as Pearson's correlation (2–3 cells per experiment, from two independent experiments). Statistical significance was determined using an unpaired *t*-test. Bars indicate the mean, and error bars represent the SEM.

G–I   The extent of colocalisation between GFP-Tollip and LAMP1 in Tom1 KO (G), ATG5 KO (H) and VPS35 KO (I) compared with wild-type (WT) HeLa cells was quantified from immunofluorescence images immunostained for GFP and LAMP1 and represented as Pearson's correlation (2–3 cells per experiment, from three independent experiments). Statistical significance was determined using a two-way ANOVA (Sidak). Bars indicate the mean, and error bars represent SEM. Western blot analysis of lysates harvested from wild type (WT) alongside either Tom1 KO (G) or VPS35 KO (I) HeLa cells was performed using antibodies specific to the indicated proteins.

Data information: Images were captured at 100× magnification. Scale bars represent 10 μm.
Source data are available online for this figure.

---

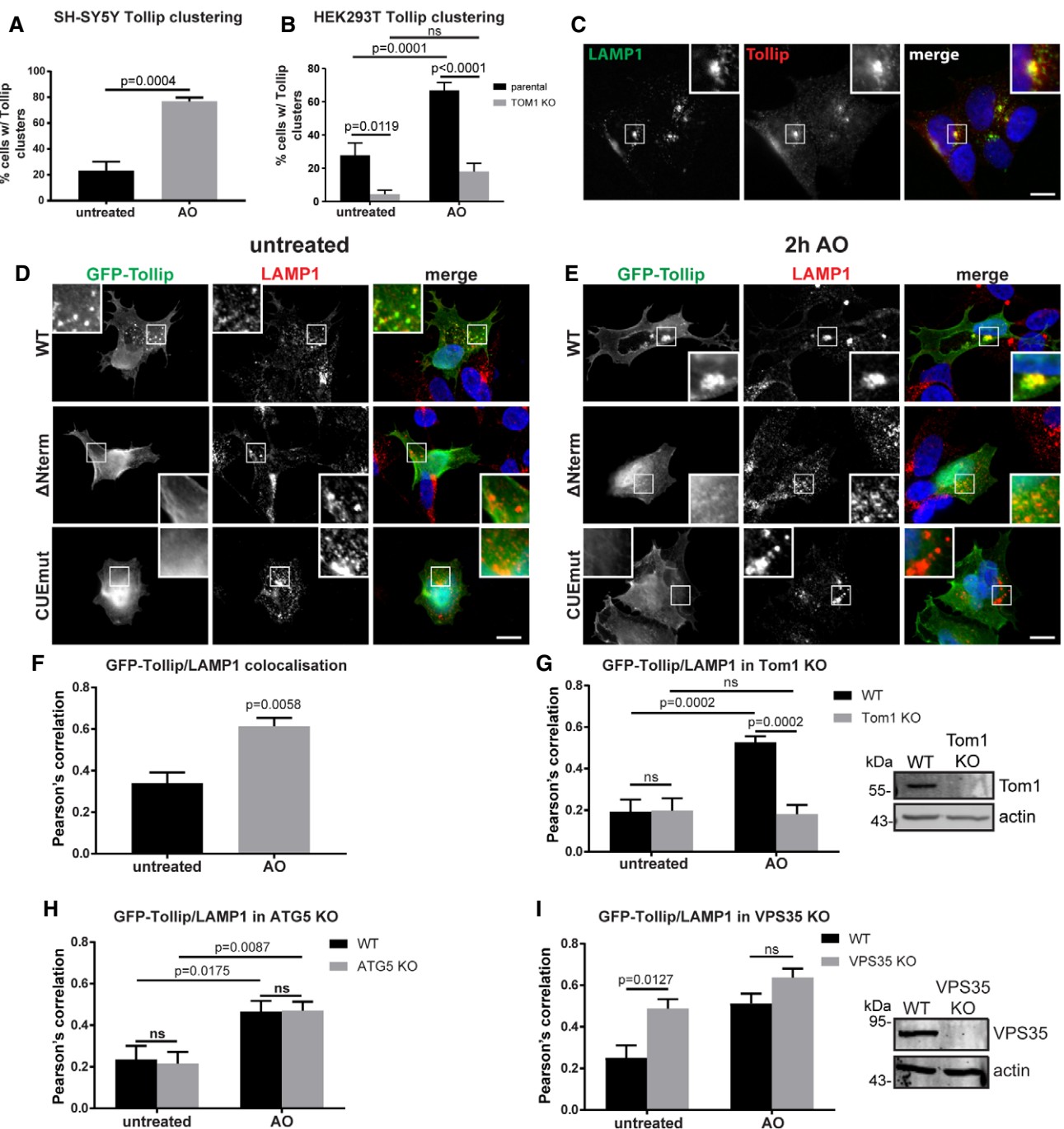

**Figure 6.**

block Tollip translocation to the LAMP1 compartment, but instead increased the level of Tollip present on this compartment (Figs 6I and EV5D). This was reminiscent of a previously observed phenotype indicating the accumulation of Rab7 on a late endosome/lysosome compartment in VPS35-depleted cells (Jimenez-Orgaz *et al*, 2018), which we replicated in our model system (Fig EV5E and F). Moreover, VPS35 knockout did not perturb the interaction of Tollip with Parkin (Figs 6I and EV5G). Together, these data would suggest that retromer is not required for Tollip regulation of Parkin-dependent MDV trafficking to the lysosome, but instead may mediate the

retrograde trafficking of Tollip away from the lysosome following its translocation under stress conditions.

### Tollip is required for MDV trafficking to a LAMP1 compartment, and this function requires Tom1, ubiquitin and membrane binding

To determine whether Tollip is essential for facilitating the trafficking of TOM20 MDVs to the lysosome to facilitate degradation, we performed siRNA knockdown of Tollip prior to the induction of

mitochondrial stress in order to analyse MDV/LAMP1 colocalisation (Fig 7A). We found that loss of Tollip reduced the level of TOM20 MDV colocalisation with LAMP1 (Fig 7B). In mock-treated cells, treatment with AO or 25 μM AA resulted in increased colocalisation of TOM20 MDVs with LAMP1 ($P = 0.0450$ and $P = 0.0002$, respectively) compared with untreated cells. This increase in association of TOM20 MDVs with LAMP1 was inhibited in Tollip siRNA cells, indicating an essential role for Tollip during MDV trafficking. This

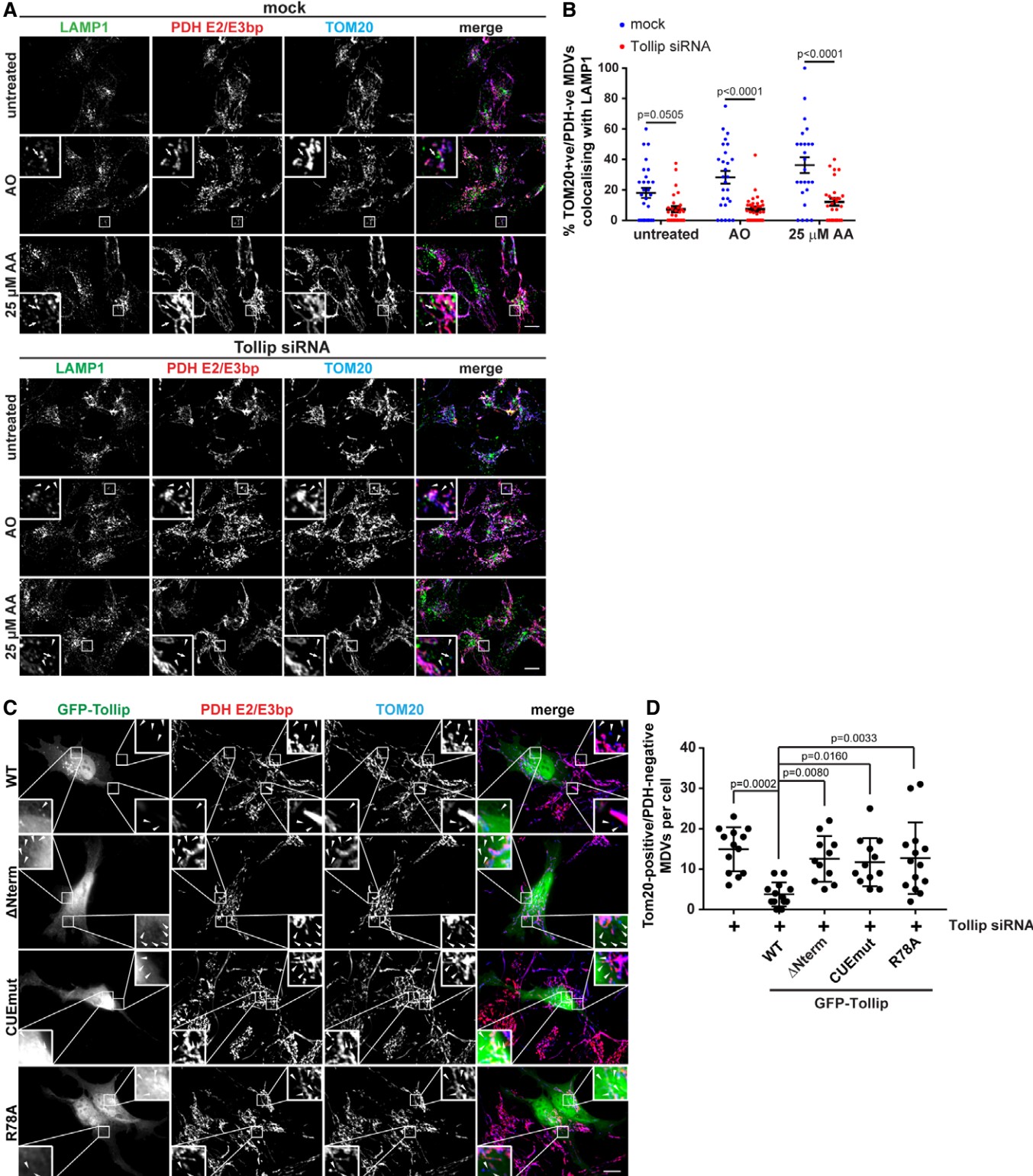

**Figure 7.**

**Figure 7. Trafficking of MDVs to the lysosome is dependent on Tollip.**

A   A subset of TOM20$^{+ve}$/PDH$^{-ve}$ MDVs are trafficked to the lysosome. siRNA knockdown of Tollip was conducted in SH-SY5Y cells over a 72-h period prior to treatment with 5 μM antimycin A/10 μM oligomycin (AO) or 25 μM antimycin A (AA) for 2 h. Cells were then fixed and stained with antibodies specific to LAMP1 (green), PDH E2/E3 bp (red) and TOM20 (blue). Arrows denote TOM20$^{+ve}$/PDH$^{-ve}$ MDVs colocalising with LAMP1, whereas arrowheads denote MDVs that do not colocalise with LAMP1. Zoom insets represent 4× magnification.

B   From immunofluorescence images, the total number of TOM20$^{+ve}$/PDH$^{-ve}$ MDVs was counted as well as the number of TOM20 MDVs that colocalised with LAMP1 to calculate the % of TOM20-positive MDVs colocalising with LAMP1 per cell ($n$ = 25–32 cells per condition from three independent experiments). Each data point represents 1 cell, and centre line indicates the mean. Statistical significance was determined using a two-way ANOVA (Dunnett). Error bars represent the SEM.

C   siRNA knockdown was performed in SH-SY5Y cells, and then 18 h prior to fixation, cells were transfected with GFP-Tollip wild type (WT), △Nterm, CUEmut or R78A. Cells were fixed and immunostained for PDH E2/E3 bp (red) and TOM20 (blue). Arrowheads denote TOM20$^{+ve}$/PDH$^{-ve}$ MDVs. Zoom insets represent 3.5× magnification.

D   Quantification of TOM20 MDVs per cell was performed by eye ($n$ = 11–14 cells per mutant from two independent experiments). Cells expressing a GFP-Tollip construct were identified using the GFP channel. Surrounding cells not expressing GFP were counted as the Tollip siRNA only population. Statistical significance was determined using a one-way ANOVA (Tukey). Each data point represents 1 cell, centre line indicates the mean, and error bars represent SEM.

Data information: Images were captured at 100× magnification. Scale bars represent 10 μm.

phenocopied observations in Parkin KO cells (Fig 4E–G), further supporting a role for Tollip and Parkin in TOM20-positive MDV targeting to the lysosome. Importantly, we were able to rescue MDV trafficking in Tollip siRNA-treated cells following the forced expression of GFP-labelled Tollip, indicated by reduced TOM20 MDVs (Fig 7C and D). Moreover, this was dependent on a fully functional Tollip, since either Tom1, ubiquitin or lipid binding was required for the rescue of MDV trafficking (Fig 7D). Since the R78A mutant is able to recruit Parkin to a vesicular compartment (Fig 2), these data would suggest that Tollip functions following MDV formation and subsequent "budding off" from the mitochondria, in order to coordinate trafficking towards the lysosomal compartment potentially via a cargo to membrane switch.

## Discussion

Our data implicate the endosomal cargo sorting protein Tollip as an essential organiser of MDV trafficking. In addition, we identify Tollip in complex with Parkin whereby it colocalises on an endosomal compartment. In the pathway described here, Tollip and Parkin function together to transport a specific subset of mitochondrial cargo to lysosomes for proteolytic turnover in response to mitochondrial insult. This identification of a Tollip/Parkin axis in MDV trafficking elucidates several key mechanisms regulating mitochondrial quality control, which may suggest further unidentified Parkin substrates are modulated at more discrete levels compared with previously defined targets by cargo-selective mechanisms to facilitate their trafficking to the lysosome.

Although Parkin is generally associated with regulating mitochondrial quality control pathways, in particular mitophagy, alternative roles exist that impact on apoptotic signalling (Carroll *et al*, 2014; Zhang *et al*, 2014; Ryan *et al*, 2018; Bernardini *et al*, 2019) and endosomal cargo trafficking (Fallon *et al*, 2006). Despite being distinct from canonical mitophagy, MDV formation, at least for some cargoes, is PINK1/Parkin-dependent (McLelland *et al*, 2014). This was determined using Parkin overexpression models. Here, we demonstrate that these MDV pathways occur in the presence of endogenous Parkin in SH-SY5Ys, indicating these mechanisms exist under physiologically relevant conditions and not just in a model that is biased to certain pathways due to excessive Parkin levels. In contrast to previous studies that used Parkin overexpression models (McLelland *et al*, 2014), we found that endogenous Parkin is present

on TOM20-positive MDVs and the production of these specific MDVs is induced upon treatment with mitochondrial stressors.

Surprisingly, Parkin depletion as well as genetic knockout led to a selective increase in TOM20 MDVs, which are unable to efficiently traffic to a LAMP1-positive compartment, suggesting a block in MDV trafficking. This contrasts with our own data as well as previous studies evaluating inner membrane-derived MDVs, such as PDH-positive, which indicate Parkin is required for MDV formation (McLelland *et al*, 2014). However, the presence of TOM20-positive MDVs in cells expressing Parkin T240R or C431S mutants would indicate that their generation is independent of Parkin E3 activity. The increase observed following Parkin knockdown and knockout along with its interaction with Tollip may suggest that the transient localisation of Parkin to these MDVs is required to facilitate their trafficking along the endosomal pathway, thus explaining why Parkin re-introduction in HeLa cells reduces the number of MDVs present in cells undergoing mitochondrial stress. Our data collectively suggest there likely exist distinct, cargo-specific mechanisms of MDV trafficking, that while may be universally Parkin-dependent, require Parkin function at distinct stages to ensure damaged mitochondrial cargo is trafficked efficiently to the lysosome. Therefore, the mechanisms of their formation and trafficking likely depend on the origin of the damaged cargo and therefore the topology of the membrane, thus requiring a distinct set of machinery.

The movement of MDVs into the endosomal system is supported by previous studies illustrating their localisation to multivesicular bodies (MVBs) (Soubannier *et al*, 2012) and their Syntaxin-17-dependent delivery to the late endosome/lysosome compartment (McLelland *et al*, 2016). Here, we found that TOM20-positive MDVs are taken up by endocytic vesicles after their "budding off" from mitochondria, as indicated by their localisation to PI(3)P-positive and EEA1-positive compartments. Interestingly, Tollip knockdown blocked this process whereas Parkin knockdown did not, despite both knockdowns resulting in MDV accumulation. This would suggest that Tollip functions to facilitate MDV transition into the endosomal system prior to Parkin activity. Tollip has also been shown to facilitate endosomal organisation and cargo trafficking, dependent on an association with the ubiquitylated adaptor p62 and the E3 ligase RNF26 (Jongsma *et al*, 2016), indicating a potentially important function for Tollip in directing ubiquitylation of E3 substrates or their subsequent trafficking. Our data indicate that a Tollip/Parkin interaction is dependent on the Tollip CUE domain,

but is not perturbed by a defect in Parkin ligase activity or mitochondrial recruitment. Together, these data would suggest that Tollip is required for MDV acquisition of PI(3)P membrane, while Parkin is recruited to MDVs to facilitate lysosomal targeting potentially through a ubiquitin-dependent process, as described for its role in regulating Rab7 activity to influence endosomal maturation (Song *et al*, 2016). The Tollip–Parkin role in MDV trafficking may be related to Parkin's function in regulating the Eps15-dependent transport of endosomal cargo (Fallon *et al*, 2006) and to its

association with the proteasomal ubiquitin receptor Rpn13 (Aguileta *et al*, 2015), via interactions mediated by its UBL domain.

In addition to Tollip/Parkin vesicles being ubiquitin-positive, a subset of these were also positive for Rab7. The increase in Tollip/Rab7 and Tollip/LAMP1 colocalisation following mitochondrial damage could suggest that Tollip shuttles MDVs through late endosomes/MVBs to lysosomes for proteolytic turnover. Indeed, Rab7 is important in late endosome/lysosome fusion (Vanlandingham & Ceresa, 2009), with its localisation regulated by retromer to facilitate Parkin-dependent mitophagy (Jimenez-Orgaz *et al*, 2018). In turn,

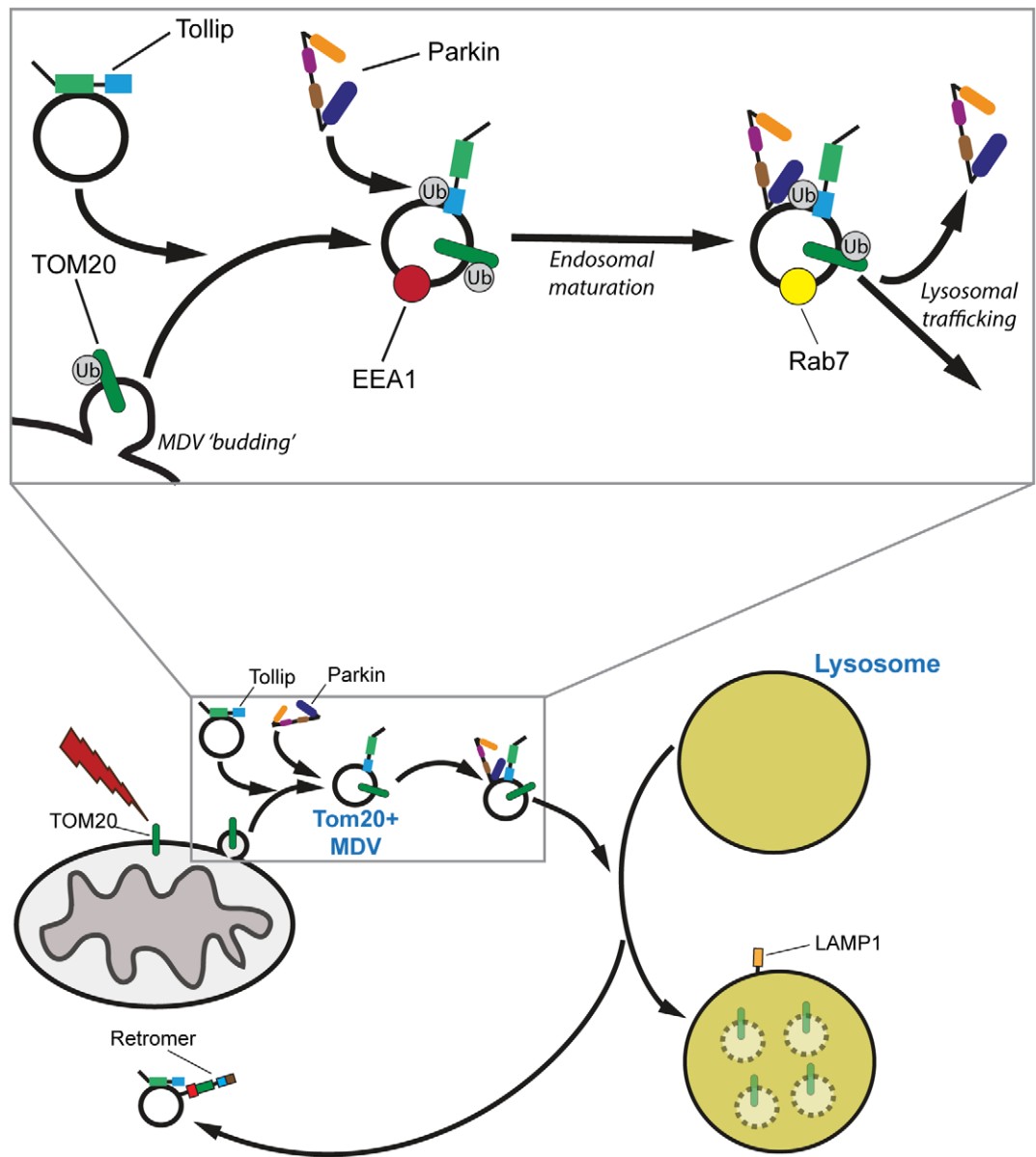

**Figure 8. Tollip and Parkin facilitate TOM20 MDV trafficking to the lysosome.**

Mitochondrial stress induces the formation of single-membrane TOM20 mitochondrial-derived vesicles (MDVs) through a currently undefined mechanism. Following budding, Tollip is essential for MDV transition into an early endosomal compartment followed by the concomitant recruitment of Parkin. Tollip and Parkin work in close association to transfer MDV cargo through the endosomal system, thus requiring both of their function for trafficking onwards to the lysosome. Fusion with endosomal intermediates to facilitate maturation and subsequently fusion with lysosomal compartments are required to mediate turnover of MDV cargo. Constituents such as Tollip and Rab7 are "recycled" from the lysosome back to endosomal bodies via a retromer-dependent pathway.

Parkin modulates the localisation and activity of both Rab7 and VPS35 (Song *et al*, 2016).

Notably, the PD-linked protein VPS35 (Vilarino-Guell *et al*, 2011; Zimprich *et al*, 2011) has been directly linked to MDV-mediated degradation pathways (Braschi *et al*, 2010) and is in fact required for the trafficking of mitochondrial dynamin-like protein 1 (DLP1) to the lysosome via TOM20-positive MDVs (Wang *et al*, 2016, 2017). Furthermore, knockout of VPS35 leads to the accumulation of overactive Rab7 on lysosomes (Jimenez-Orgaz *et al*, 2018), thus perturbing Rab7 function. Interestingly, we observed lysosomal accumulation of Tollip in VPS35 knockout cells, suggesting retromer may regulate Tollip's spatiotemporal function. Similar to Tollip knockdown data shown here, depletion of VPS35 leads to an increase in TOM20-positive MDVs (Braschi *et al*, 2010), supporting a hypothesis whereby both proteins operate synergistically along the same MDV pathway. With an association between Tollip and Rab5 membrane also being observed in cells under basal conditions, it is possible that Tollip utilises previously defined endosomal pathways (Rink *et al*, 2005; Poteryaev *et al*, 2010) to initiate Rab7-dependent trafficking of MDVs upon mitochondrial damage. This observation was also interesting as Rab5 mediates a Parkin-dependent mitochondrial clearance pathway (Hammerling *et al*, 2017a), further highlighting the importance of endosomal organisation in mitostasis.

It is important to note that Tollip was first identified as a negative regulator of interleukin-1 and Toll-like receptor signalling to modulate the innate immune response (Burns *et al*, 2000; Zhang *et al*, 2014). Interestingly, both PINK1 and Parkin have recently been suggested to be suppressors of an immune response pathway downstream of mitochondrial damage. The absence of Parkin function in the context of inflammation induces mitochondrial antigen presentation via a MDV pathway (Matheoud *et al*, 2016). In addition, experiments performed in Parkin null mice exposed to exhaustive exercise indicate excessive inflammation as a result of STING-dependent pro-inflammatory cytokine production (Sliter *et al*, 2018). In relation to this, there is very recent evidence suggesting Tollip directly regulates STING activity (Pokatayev *et al*, 2020), suggesting that sequestration of Tollip towards MDVs in response to mitochondrial damage may dampen the immune response, similar to the effect described for Parkin (Sliter *et al*, 2018). Therefore, it remains to be determined whether Tollip regulation of a discrete MDV pathway plays an important role to regulate the innate immune response downstream of mitochondrial-induced damage.

In conclusion, we suggest a model whereby TOM20-positive MDVs "bud off" from the mitochondria and are then "captured" by Tollip-positive endosomes, which facilitate MDV trafficking to the lysosome (Fig 8), consistent with Tollip's role as an organiser of the endosomal network (Jongsma *et al*, 2016). During the transition of an MDV into a PI(3)P rich endosomal compartment, Tollip may recruit a cytosolic or endosomal pool of Parkin by direct or secondary associations, but dependent on CUE domain interactions. Alternatively, due to Parkin's rapid accumulation on mitochondria in response to damage, it may act to recruit a Tollip-associated membrane compartment to facilitate the transition of damaged cargo into the endosomal system. Parkin then likely functions on these mitochondrial cargo-containing endosomes to facilitate trafficking to the lysosome in a subsequent ubiquitin-dependent or ubiquitin-independent step. Following this, retromer functions to

either directly, or via Rab7-positive compartments, traffic Tollip from lysosomes back to early endosomes. Overall, these selective mitochondrial quality control pathways may represent a more physiologically relevant mechanism in PD pathology compared with mitophagy, and thus, dissecting their distinct mechanisms may provide novel and effective therapeutic targets.

# Materials and Methods

### Antibodies and reagents

Primary antibodies used for Western blotting were specific against Tollip (GTX116566, 1:1,000) from GeneTex; Tom1 (ab99356, 1:1,000), Parkin (ab77924, 1:500), COXII (MTCO2, ab110258, 1:1,000), PDH E2/E3bp (ab110333, 1:1,000) and Rab7a (ab137029, 1:500) from Abcam; Actin (612656, 1:2,000) and LAMP1 (555798, 1:500) from BD Biosciences; GAPDH (10494-1-AP, 1:2,000) from ProteinTech; MFN2 (7581, 1:500), PINK1 (6946, 1:500) and ATG5 (26305, 1:500) from Cell Signalling; TOM20 (SC-11415, 1:2,000) and VPS35 (SC-374372, 1:1,000) from Santa Cruz; GFP (A11122, 1:1,000) from Invitrogen; Myc (9E10, MAB3696-SP, 1:1,000) and K48-linked ubiquitin (A-101, 1:20,000) from R & D Systems; and ubiquitin (FK2, BML-PW8810-0100, 1:2,000) from Enzo. Primary antibodies used from immunofluorescence were specific against Tollip (GTX116566, 1:100) from GeneTex or (ATO0918, 1:100) from Insight Biotechnology Ltd; Tom1 (ab99356, 1:100), Parkin (ab77924, 1:250), PDH E2/E3bp (ab110333, 1:1,000), Rab7a (ab137029, 1:100) and GFP (A11122, 1:1,000) from Abcam; LAMP1 (555798, 1:250), Rab5 (610282, 1:200), EEA1 (610456, 1:250) and GM130 (610823, 1:1,000) from BD Biosciences; HA (901501, 1:500) and Cytochrome c (612302, 1:1,000) from BioLegend; MFN2 (7581, 1:500) and HA (37245, 1:1,000) from Cell Signalling; TOM20 (SC-11415, 1:1,000) and VPS35 (SC-374372, 1:1,000) from Santa Cruz; GFP (A11122, 1:2,000) from Invitrogen; Myc (9E10, MAB3696-SP, 1:1,000) from R & D Systems; ubiquitin (FK2, BML-PW8810-0100, 1:1,000) from Enzo; and Cathepsin D (IM16, 1:100) from Oncogene.

Secondary antibodies against mouse (680) and rabbit (800) raised in goat for Western blotting were sourced from LI-COR Biosciences and used at 1:5,000 in TBS (0.01% SDS). Secondary antibodies used for immunofluorescence were all raised in goat and sourced from Thermo Fisher Scientific. These were Alexa Fluor anti-rabbit 488 (A11034), anti-mouse 488 (A11029), anti-mouse IgG1 488 (A21121), anti-mouse 568 (A11031), anti-rabbit 568 (A11036), anti-mouse IgG2b 568 (A21144), anti-mouse IgG2b 594 (A21145), anti-rabbit 633 (A21070) and anti-mouse IgG1 647 (A21240). Hoechst (bisbenzimide) for nuclei staining was sourced from Sigma.

The following reagents were used: bafilomycin A1 (328120001) from Acros Organics, oligomycin (495455) from Millipore and antimycin A (A8674) from Sigma-Aldrich.

### Plasmids

pcDNA3.1 mycBioID was a gift from Kyle Roux (Addgene plasmid #35700). The mycBioID cassette was then PCR-subcloned via AgeI and XhoI into pEGFPc1 in place of the eGFP cassette (pmyc-BioIDc1). mycBioID was cut and pasted into pIRESneo2 using

AgeI and BamHI. pCMV6-AC Tollip was purchased from OriGene (SC320186), and Tollip was cut and pasted into pEGFPc3 via HindIII and EcoRI. Tollip was PCR-subcloned into pmycBioIDc1 via HindIII/EcoRI and subcloned into pIRESneo2 using NheI and EcoRI. The Tollip R78A mutation was made via PCR primers that contained a double nucleotide substitution to GC at position 232 and position 233. The Tollip CUEmut (M240A/F241A) was made using PCR primers to substitute a CG at position 718 and position 719, as well as a GC substitution at position 721 and position 722. The Tollip F21A mutation was made via PCR using primers that substituted a GC at position 21 and position 22. The Tollip ΔNterm was created by truncation of the first 53 amino acids and PCR-subcloned into pEGFPc3 via HindIII/EcoRI. pRK5-HA-Parkin was a gift from Ted Dawson (Addgene plasmid #17613), and HA-Parkin was cut and pasted with EcoRI/NotI into pIRESpuro2. The Parkin R42P mutant was made using PCR primers to substitute a G nucleotide at position 125 to a C nucleotide. For the Parkin RING1 domain mutants T240R and R275W, PCR primers were used to substitute nucleotides C at positions 719 and 823 to a G and T, respectively. The Parkin C431S mutant was made using PCR primers to substitute a G nucleotide at position 1292 to a C nucleotide. GFP-Rab7 was a kind gift from Matthew Seaman (CIMR, University of Cambridge), and GFP-Rab5 was a kind gift from Folma Buss (CIMR, University of Cambridge). GFP-2xFYVE in a pEGFPc3 vector was a kind gift from Prof Nullin Divecha (University of Southampton).

## Cell culture and transfection

SH-SY5Y cells were maintained in DMEM/F-12, GlutaMAX™ (Gibco) supplemented with 1× non-essential amino acids (NEAAs; Gibco) and 10% FCS (Gibco). HeLa cells were maintained in RPMI media (Gibco) supplemented with 10% FCS. HEK293 cells were maintained in DMEM (Gibco) supplemented with 10% FCS. Transfections were performed using FuGENE® 6 Reagent (Promega) according to the manufacturer's instructions. For generation of stable transfected lines, cells were incubated with transfection reagent and plasmid for 24 h. Media were then removed, and cells washed with PBS prior to the addition of selection antibiotics in fresh media for 72 h.

To induce mitochondrial stress, cells were treated in complete growth media with 5 μM antimycin A and 10 μM oligomycin in combination (AO), 25 μM antimycin A (AA) or 20 μM CCCP unless otherwise stated for indicated times. Bafilomycin A1 (BfnA1) was used at 100 nM individually or in combination with other stressors.

## Generation of CRISPR/Cas9 cell lines

Tollip KO cells were generated in HEK293T cells using the gRNA sequence ACCACCGTCAGCACTCAGCG targeting exon 1 cloned into the px459 plasmid (Addgene; 62988). This was transfected with FuGENE6 (Promega), and 24 h following transfection, cells were selected in 1.5 μg/ml puromycin for 48 h prior to single-cell cloning. Clonal KO lines were screened by Western blot, and genotypes were validated by sequencing. The Tollip KO clone has a single base pair adenine insertion just upstream up the PAM site that results in a 15-amino acid truncated product as a consequence of a nonsense mutation.

Tom1 KO and Atg5 KO cells were generated in HeLa cells using a single Atg5 gRNA (AAGATGTGCTTCGAGATGTG) targeting exon 2 or two separate Tom1 gRNAs in combination (GAACCCGTTCAGCTCTCCAG and ACCGCCGCTGCCACCAACCC) targeting exon 1. These were transfected into cells with FuGENE 6 and 24 h later selected in 1.5 μg/ml of puromycin for 48 h. Cells were single-cell-cloned and screened by Western blot.

The VPS35 KO HeLa cells were created following cotransfection of 3 separate gRNAs alongside a GFP-tagged puromycin plasmid. These px330 VPS35 gRNAs were a kind gift from Florian Steinberg (Jimenez-Orgaz *et al*, 2018). The transfected cells were selected in 3 μg/ml puromycin for 24 h, followed by single-cell cloning. VPS35 KO clones were confirmed by Western blot analysis.

The Tollip and Parkin KO SH-SY5Y cells were generated by lentiviral CRISPR methods using the Tollip gRNA sequence previously described or the Parkin gRNA sequence GGTGGTTGCTAAGCGACAGG targeting exon 2 cloned into the BsmBI digested LentiCRISPRv1 plasmid (Addgene; 49535). Lentivirus was produced in HEK293FT cells using 3rd-generation lentiviral packaging vectors. Viral supernatants were collected and used to infect SH-SY5Y cells in the presence of 5 μg/ml polybrene. After 24 h, viral supernatant was washed out and cells were allowed to recover in fresh growth media for an additional 24 h, prior to selection in 1.5 μg/ml puromycin. Heterogeneous populations were screened for loss of protein by Western blot before being used for relevant experiments.

## siRNA knockdowns

siRNA-mediated knockdown of target genes was performed in SH-SY5Y cells. Specific siRNA oligos against Tollip (s29037), Tom1 (s19514) and Parkin (s10043) were purchased from Life Technologies. Prior to seeding cells, transfection mix containing Lipofectamine RNAiMAX Reagent (Life Technologies) and specific siRNA oligo in Opti-MEM-reduced serum medium was made up according to manufacturer's instructions. Mock-treated cells lacked siRNA oligo and were therefore incubated with transfection reagent alone. Reverse transfection was performed by seeding cells onto transfection mix, with a final siRNA concentration of 50 nM, and incubated for 24 h. Media were then removed, and fresh transfection mix in cell media was then added at the same concentrations and cells incubated for a further 48 h.

## Protein extraction and Western blotting

Prior to lysis, cells were gently washed with ice-cold PBS and then lysed with SDS–Triton lysis buffer on ice and scraped. Lysates were centrifuged at 11,000× *g* for 10 min at 4°C and then supernatant removed. Protein concentrations were then calculated using a Pierce™ BCA Protein Assay Kit (Thermo Scientific), according to manufacturer's instructions. Samples were then diluted and mixed with an equal amount of 2× SDS loading buffer.

For Western blot analyses, samples were heated to 95°C for 5 min, and an equal amount of protein loaded per well and then separated on SDS–PAGE denaturing gels and transferred onto PVDF membranes. Membranes were blocked in 5% milk for 1 h, prior to incubation overnight at 4°C with primary antibodies. Membranes were washed in TBS-T 3 times then incubated for 1 h with fluorescently labelled LI-COR secondary antibodies and visualised using a LI-COR imaging system.

Quantification was performed by densitometry using Image Studio Lite software and samples normalised to loading controls.

### ATP assay

SH-SY5Y cells were plated in 96-well plates at a density of 40,000 cells/well in replicate wells for each condition. Replicate plates were utilised to allow for measurement of both ATP production and total protein. Following an overnight recovery, cells were washed and replenished with DMEM without glucose (Thermo Fisher Scientific, 11966025) containing 10 mM galactose or with normal growth media containing glucose, in the absence or presence of 10 μM oligomycin. Cells were incubated for 2 h at 37°C prior to harvesting for ATP production using the Mitochondrial ToxGlo Assay (Promega, G8000) or for protein using a BCA assay. For measuring ATP, luminescence readings were captured on a GloMax Microplate Reader (Promega). A background subtraction was performed (media only) on these values and normalised against the total protein content as measured by a BCA protein assay. Results represent replicate readings, across 3–4 independent experiments for each condition.

### Mitochondrial isolation

HEK293 cells were grown in 100-mm dishes to approximately 80% confluence, with 3 dishes used per condition. Cells were treated with AO for 2 h, then media removed and cells gently washed in PBS twice. Mitochondria were then purified using a Mitochondrial Isolation Kit (Abcam, ab110170) according to manufacturer's instructions. Samples were then analysed by SDS–PAGE and Western blotting, as described.

### BioID assays

HeLa WT or knockout cell lines were transfected with a Myc-tagged BioID-Tollip construct, or empty vector, then selected using 500 μg/ml Geneticin® and clonal colonies isolated and screened by Western blot analysis for expression of the construct. Clonal lines were then transfected with a HA-Parkin construct and selected using 1.5 μg/ml puromycin.

   BioID cell lines expressing HA-Parkin were then seeded in 100-mm dishes in DMEM 24 h prior to treatment, when fresh DMEM containing 50 μM biotin and stressor (or vehicle) was added for 6 or 24 h. A biotin-free condition was used to assess background. Media were then removed, cells washed twice in ice-cold PBS and 500 μl of cold lysis buffer (500 mM NaCl, 0.4% SDS, 2% Triton X-100, 5 mM EDTA, 1 mM DTT in 50 mM Tris–HCl pH 7.4 and 1× cOmplete™ protease inhibitor cocktail) added before scraping cells. Lysates were mixed with an equal amount of 50 mM Tris–HCl pH 7.4, then centrifuged at 11,000× $g$ for 15 min at 4°C and supernatant transferred onto streptavidin–agarose beads. A small amount of supernatant was kept for whole-cell lysates. Pulldowns were performed on streptavidin–agarose beads on a rotator overnight at 4°C. For washing, beads were pelleted at 11,000× $g$ for 1 min at 4°C and resuspended in wash buffer (50 mM Tris pH 7.4, 250 mM NaCl, 0.2% SDS, 1% Triton X-100, 2.5 mM EDTA, 0.5 mM DTT) four times. After last wash step, a small amount of liquid was left on top of beads and 2× SDS loading buffer added. Samples were boiled and analysed by Western blotting.

### Co-immunoprecipitation and ubiquitin-binding assay

For coimmunoprecipitations, HeLa cells stably coexpressing GFP-Tollip and HA-Parkin or transiently transfected with GFP-Tollip wild type and mutants were lysed in 50 mM Tris pH 7.4, 100 mM NaCl, 10% glycerol, 1% NP-40, 5 mM MgCl$_2$ and cOmplete™ protease inhibitor cocktail (Sigma-Aldrich). Lysates were centrifuged at 11,000× $g$ for 10 min at 4°C, and the supernatant was collected. 30 μl of lysate was taken out for lysate input and the rest was incubated with either 0.35 μg of normal rabbit IgG (Sigma-Aldrich) or anti-HA antibody (Cell Signalling), or 5 μg of anti-GFP antibody for 2 h at 4°C, rotating. Following this, 15 μl of protein A/G agarose (2BScientific; X1205) was added and incubated for 1 h at 4°C, rotating. Beads were washed 4×, 1 ml with lysis buffer followed by addition of 2× SDS sample buffer to both the lysates and beads. Samples were boiled for 5 min, prior to loading on 7.5% SDS–PAGE.

   For ubiquitin-binding assays, HeLa cells expressing either GFP alone or GFP-Tollip wild type and mutants were lysed in pulldown buffer (50 mM Tris pH 7.4, 150 mM NaCl, 0.2% Triton X-100, 1 mM EDTA and cOmplete™ protease inhibitor cocktail). Lysates were centrifuged at 11,000× $g$ for 10 min at 4°C, and supernatant was collected. The lysate supernatant was incubated with 5 μg of anti-GFP antibody for 2 h at 4°C, rotating. Following this, 20 μl of protein A/G agarose was added and samples incubated for 1 h at 4°C, rotating. Beads were washed 3×, 1 ml with pulldown buffer. After the last wash, the volume was brought up to 600 μl with pulldown buffer and 0.5 μl (2.5 μg) K63-linked polyubiquitin/Ub2-Ub7 (UC-330; Boston Biochem) was added. These samples were incubated for 1.5 h at 4°C, rotating. Beads were washed 4×, 1 ml with pulldown buffer followed by addition of 2× SDS sample buffer. Samples were run on 15% SDS–PAGE and processed for Western blot analysis. Membranes were probed with anti-ubiquitin P4D1 monoclonal antibody (eBioscience) and anti-GFP rabbit polyclonal antibody (Life technologies).

### Immunofluorescence microscopy

Cells cultured on MeOH-sterilised glass coverslips were fixed in 4 or 6% paraformaldehyde followed by permeabilisation in 0.1% Tween-20 in PBS for 5 min. After washing in PBS, coverslips were incubated in blocking buffer (1% BSA in PBS) for 15 min prior to incubation with primary antibodies in blocking buffer for 1 h. Coverslips were then washed three times in PBS and incubated with appropriate Alexa Fluor-conjugated secondary antibodies in blocking solution for 1 h in the dark. After 3× PBS washes, coverslips were washed with dH$_2$O and then mounted onto glass slides using FluorSave™ Reagent (Merck Millipore) overnight. Images were captured on a Zeiss Axioplan 2 microscope coupled with ImageJ Fiji software. When required, background was subtracted from individual channel images using the rolling ball method within Fiji software prior to image thresholding. Super-resolution radial fluctuation (SRRF) imaging was performed on a Nikon Eclipse Ti microscope coupled with Fiji software. To analyse data and generate images, the NanoJ-SRRF ImageJ plugin was used (Gustafsson *et al*, 2016), with time-lapse imaging used to capture 100 frames per channel.

   MDVs were defined using 2 individual mitochondrial antibodies in order to identify vesicles containing discrete cargo and analysis performed by eye. ROI Manager in ImageJ Fiji software was used to

quantify MDVs containing one mitochondrial marker and lacking another, as well as identifying their colocalisation with other compartments. Where required, colocalisation analysis between two channels was performed using the Colocalisation Test Fiji plugin, which utilises the Costes method (Costes *et al*, 2004), to calculate Pearson's correlation.

## Statistics

All graph production and statistics were performed using GraphPad Prism 7 software. All error bars represent SEM. Two-tailed unpaired *t*-tests were used for statistical analysis between two sample groups. One-way or two-way ANOVA tests coupled with multiple comparisons *post hoc* testing were used for statistical analysis between multiple groups, as appropriate. All $P < 0.05$ were considered statistically significant.

# Data availability

The data that support the findings of this study are available from the authors upon reasonable request.

**Expanded View** for this article is available online.

## Acknowledgements

We would like to thank Professor Nullin Divecha for the GFP-FYVE plasmid and Dr Florian Steinberg for the VPS35 CRISPR targeting plasmids. In addition, we thank Dr Mark Willett for assistance in the Biological Sciences imaging and microscopy centre. TAR was supported by a Wellcome Trust Seed Award to DAT (205909/Z/17/Z). EAA was supported by a Gerald Kerkut Trust PhD studentship. CLC was supported by a Wessex Medical Research Trust PhD studentship.

## Author contributions

TAR designed and performed a majority of the experiments, analysed the data and wrote the manuscript. DAT conceived the study, designed and performed some of the experiments, analysed the data and edited the manuscript. EOP generated the VPS35 KO data, AJBR developed the BioID–Tollip assay, DR evaluated Tollip subcellular localisation, CLC generated SH-SY5Y Tollip and Parkin KO data, REW generated the ATP data, and EAA generated the Atg5 KO cell lines.

## Conflict of interest

The authors declare that they have no conflict of interest.

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
