## [Review Process File · The EMBO Journal]

Tollip coordinates Parkin-dependent trafficking of mitochondria-derived vesicles

Thomas Ryan, Elliott Phillips, Charlotte Collier, Alice Robinson, Daniel Routledge, Rebecca Wood, Emelia Assar and David Tumbarello.

Review timeline:

Submission date:	31 st May 2019
Editorial Decision:	18 th July 2019
Revision received:	24 th January 2020
Editorial Decision:	21 st February 2020
Revision received:	28 th February 2020
Accepted:	10 th March 2020

Editor: Elisabetta Argenzio

Transaction Report:

1st Editorial Decision

18th July 2019

Thank you for submitting your manuscript entitled "Tollip coordinates Parkin-dependent mitochondrial derived vesicle trafficking" [EMBOJ-2019-102539] to The EMBO Journal. I have now read your letter and discussed it with the other members of our editorial team.

The consensus is that the plan to address the referees' concerns appears reasonable. Given the overall interest of your study, I am pleased to invite submission of a revised manuscript as indicated in the referee's reports and characterizes the Tollip-Parkin interaction at both mechanistic and physiological levels. I would like to point it out that addressing all the referees' points in a conclusive manner will be essential for publication in The EMBO Journal, as well as a strong support from the referees.

REFEREE REPORTS

Referee #1:

In this study by Ryan et al., the authors identify a role for the Toll-interacting protein Tollip in the targeting of Tom20+ mitochondrial derived vesicles to the endosomal compartment. This occurs through its interaction with Parkin after the vesicles have formed, and appears independent of Parkin's ubiquitination E3 ligase activity, but dependent upon Tollip's ubiquitin binding CUE domain. Loss of Tollip leads to an accumulation of MDVs stimulated with oxidative stress, consistent with a requirement for this protein in later steps of vesicle targeting. The manuscript defines Tollip primarily as an adaptor to sort ubiquitinated receptors through the endocytic compartment, which is of course correct, but it should be noted that this protein is primarily studied

as a central adaptor in innate immune signaling (hence the name). Given the rapid emergence of mitochondria as a signaling platform in innate and adaptive immunity, the impact of this study is likely to be much broader than advertised here in terms of quality control. Both the McBride/Desjardins (Cell 2016) and Youle (Nature 2017) labs recently highlighted the role of Pink1 and Parkin as repressors of innate immunity, although they apparently disagree on the mechanisms. I say this not in request of experiments along this line, but to state my enthusiasm for the future implications of this work. The authors should at least comment on the primary function of Tollip in immunity to alert the readers to the majority of the literature on this protein.

The authors challenge the cells with antimycin/oligomycin (AO) or antimycin alone for a few hours to generate ROS-induced MDVs, and they focus on this in the context of protein quality control. This work differs from previous work analyzing Tom20+ MDVs, which were initially shown to be independent of PINK1/Parkin. In agreement, they show that Tom20 MDV formation does not require Parkin, but that they accumulate. How exactly Tollip and Parkin work together to drive these vesicles to the lysosome is investigated, but the overall conclusions are rather confusing and speculation tends to over-step the data.

Comments and suggestions

- Fig 1G shows Tom20 and COXII protein levels reduced by half in control cells treated with AO for 24 hours, and that only the loss of Tom20 was dependent upon Tollip (Fig 1G). The blots shown are not convincing here, and a 24 hour time point would almost certainly invoke mitophagy and other pathways as the damage accumulates. What do the cells and mitochondria look like at this time point? Have the authors looked at any differences when cells are adapted to galactose rather than in high glucose? In general, it would be helpful if the authors performed this experiment as a time course and probed with a number of additional mitochondrial markers to interrogate mitophagy and potentially protease (LonP) pathways. It is not yet clear what other cargoes are within Tom20 vesicles, so this could be informative regardless of the outcome. As a suggestion, other outer membrane proteins like the mitofusins or Bcl-2 family proteins, cytochrome c for the IMS and PDH should be examined to make their point more clearly. In vitro budding assays from the McBride lab quantified protein release by MDVs with stressors as containing just a few percent of total mitochondrial protein per hour. A loss of 50% of total protein in 24 hours may therefore reflect other mechanisms, including reduced translation, import, etc.

- The fluorescent imaging of GFP-Tollip with Parkin, ubiquitin, Tom20, etc., is difficult to see very clearly. There appears to be a lot of background staining throughout the cytosol, with a very high expression in the nucleus in many cases, making it difficult to interpret a few colocalizing dots. This is an issue with the study of MDVs in general, so in each case the authors must take care to validate all antibodies that are used for IF in siRNA or KO cell lines. I appreciate the use of additional cell lines in sup figures, and parkin-GFP, but it remains a concern. Silencing Drp1 would help minimize mitochondrial fragmentation and allow an easier view of MDVs carrying Tom20 or PDH as well, where applicable, particularly when only one of the two markers are being monitored (for localization with Tollip, Parkin, ub, etc).

- Do the authors see changes in PDH+ MDVs upon loss of Parkin, as previously observed? There is very little consideration of the potential differences between Tom20 and PDH MDVs and why they would transit to the late endosome in different pathways. Do they think that Tom20 MDVs target Rab5/EEA1 early endosomes and PDH directly targets multivesicular bodies?

- GFP-Tollip localizes to cytosol in untreated cells, but shows a very strong translocation to the nucleus upon treatment (see Fig 2, particularly with the mutants). Is this a real translocation, or does it reflect cleavage of the GFP during the stress?

- The data is often presented as % of Tollip (or ubiquitin, etc) puncta that are positive for Y, but how many puncta are there? How is a puncta defined? Tollip is known to act within the endosomal compartments, so how many of these "puncta" are endosomes? Some of this is shown in S6 and S7, but quantification is as a pearsons co-efficient, not as "puncta". So trying to reconcile the presentation of data between different figures is difficult.

- Loss of the lipid binding domain of Tollip increased the colocalization with Parkin, whereas loss of the ubiquitin binding CUE domain abolished this, both in fluorescence and with BioID approaches with Tollip as bait. This indicates Tollip requires ubiquitin to interact with Parkin. However, mutations in the Parkin ub ligase domain actually increased the interaction with Tollip even without AO treatment, which is lost upon treatment (Fig 4E R275W mutant in BioID). This result is not well described in the results section and is very confusing. Where are they interacting? What happens to

PDH+ MDVs in these conditions? Where is the CUE requirement coming from, if parkin ub mutants bind so much more strongly? Does the CUE mutant bind the R275E mutant?

- The authors show that loss of Parkin leads to an increase in Tom20 MDVs, consistent with previous work showing Parkin is not required for the formation of this class of vesicles. The distinction in the current study is that they observe some recruitment of Parkin to Tom20 MDVs after they form, something not observed in previous work. Unlike Tollip, the increased number of Tom20 MDVs is not due to a requirement for Parkin to target and enter the endocytic compartment, since the localization with the FYVE probe was retained. Again, this is confusing. Perhaps a likely explanation is that loss of Parkin results in increased mitochondrial damage, which increased Tom20 MDV generation and flux, without a real "block" anywhere. But in terms of the Parkin recruitment to these MDVs, if Parkin is required to recruit Tollip to the MDVs, and Tollip is required to deliver the MDVs to the endosome, why isn't Parkin required for that step as well? The authors suggest Parkin is requisite for the latter step of fusion/maturation into a late endosome, but this is not borne out from the data presented.

- When examining Tollip localization with LAMP1 the authors state "we observed an increase in the trafficking of Tollip to LAMP1 compartments (Figure 6F)". Do they mean Tollip arrived there on MDVs, or was recruited from cytosol to the late endosome? The signals for recruitment are distinct, and I am not clear what is meant by this, or whether they can interpret the colocalization as being due to traffic. There is generally a lot of speculation in the discussion about the mechanisms and steps between MDV, early and late endosome. They speculate that MDVs "piggy-back" on Tollip endosomes, which is confusing given their data showing apparent complete colocalization (as a single vesicle).

Overall, the primary conclusions are that Tom20 MDVs, but not PDH MDVs, require Tollip to transport and fuse with the endocytic compartment. Parkin interacts with Tollip on a subset of Tom20 MDVs, and its loss also increases the number of these MDVs. However, the details of how these two proteins work together, and at which level of endosome, is not well resolved at this point. The strength of the study is the novel association of this adaptor, so well studied within the immune system, with the Parkin regulated pathways of mitochondrial vesicle formation.

Referee #2:

In this manuscript the authors investigate the role of Tollip in a particular pathway linked to mitochondrial stress response and quality control. This pathway involves the formation of MDVs and its trafficking to lysosomes via the endosomal pathway and is distinct to the classical Parkin-dependent mitophagy pathway. The authors nicely address different stages and dependencies along this route. Despite the interaction of Tollip with Parkin, Tollip does not participate in the classical mitophagy pathway, but instead has an important role in the MDV trafficking pathway. The authors show that Tollip interacts with a subset of MDVs and helps in their delivery to an endosomal compartment of the cell.

In particular, the authors show that knockdown of Tollip increases the occurrence of TOM20 positive MDVs and that Tollip colocalises to some of these MDVs. They show a stress-induced colocalisation of Tollip with Parkin and a dependency of this interaction on the ubiquitin-interacting CUE domain of Tollip. This Parkin-Tollip interaction was confirmed by proximity labelling. Some of the Tom20-positive MDVs colocalised with 2xFYVE in a Tollip-dependent but Parkin-independent manner. Tollip clustering occurred at LAMP1 positive sites in the cell. This was shown to be partially dependent on the presence of TOM1 and Tollip's interaction domain with TOM1. Finally, the authors showed that some of these MDVs are also positive for LAMP1 and that this colocalisation is dependent on Tollip.

Overall, this is a very comprehensive and well-performed study given important insights into this novel pathway of mitochondrial QC. It focuses on the role of Tollip but also gives novel insights into the role Parkin in MDV trafficking and targeting to lysosomes via endosomes. A few issues still need to be addressed or clarified in my opinion as they are not clear or not convincingly shown.

Major issues:

1. The accumulation of Tom20 MDVs upon depletion of Tollip is taken an indication of Tollip playing a role in trafficking rather than playing a role in its formation. Although these is quite likely,

- still, the authors did not exclude that Tollip induces some sort of mitochondrial dysfunction that could initiate MDV formation. The authors should test whether mitochondrial respiration or morphology is altered upon depletion of Tollip or explain why they think they can exclude this.
2. Regarding the text describing figure 4 (page 9 paragraph 2): Have the authors considered that the Tollip-Parkin interaction might be mediated directly via the UBL domain of Parkin instead of ubiquitin itself? Is CUE-domain-dependent binding of Tollip occurring directly to the UBL or indirectly via ubiquitinated Parkin? This would be an important point to address or to discuss more clearly in my opinion.
 3. Along this line, how does the used R42P Parkin mutant affect MDV formation under mitochondrial stress (see Fig. 4F)? Does it also impair MDV formation?
 4. Fig. 5G: Showing the standard error of the mean (SEM) in a n=2 experiment is questionable. It's better to show the standard deviation (SD) which is equal to the range of the two experiments, or show the two values as dots and the average as a line. How can the authors do statistics on n=2 experiments? Please adapt or include more experiments.
 5. In the discussion (last paragraph) the authors might also discuss that under stress most Parkin is rapidly located at mitochondria and thus Tollip-Parkin interactions might occur here as well, not only via binding to cytosolic Parkin.

Minor issues:

6. The significance asterisks should be used uniformly throughout the manuscript. In figure 2 ** is given as $p < 0.005$ while the same p-value for figures 4, 6 and 7 is represented by ***
7. It is not entirely clear whether the quantification analyses are different for Fig 2C, G, H, I, J and 5C. The authors should explain this in more detail. See also the Comment to 5C for this. If they indeed quantified the colocalisation differently, why did they do so?
8. Fig. 1E: A quantification for the occurrence of these events would be helpful.
9. Fig. 1B, G: Was non-targeting siRNA used as negative ctrl (mock). What does 'mock' mean? Please clarify.
10. Fig. 1G, H, I: This conclusion is not convincing. It is also unlikely that the pool of Tom20 is largely affected given the fact that most of Tom20 remains on intact mitochondria and only a minor fraction is at MDVs anyway. Moreover, the example blot does not quite reflect the quantification, is there a better example? Some bands look very strong. Are the authors sure that there is no saturation, which would affect the quantification? I feel that this is overstated, but also not required. Please correct or explain.
11. Fig. 2A: How many of these events were observed? Can the authors give an estimate or a quantification for the occurrence of these events?
12. Fig. 2B, E, F: it is not clear which colour was attributed to ubiquitin in the merge picture. Was white chosen as a color? If so, this is not ideal to visualizing a colocalisation. The authors should rather use a different false colouring for ubiquitin.
13. Fig. 3: Although highly unlikely, there is the possibility that Tollip somehow interacts with the HA-tag instead of Parkin. Technically, a proper control e.g. HA-GFP to exclude this is missing.
14. Fig. 3 caption. There is a typo in the figure caption. "Tollip specifically interacts with Tollip". Please correct.
15. Fig. 4A, B, C: Was non-targeting siRNA used as negative ctrl (mock) or does mock mean no siRNA at all, just transfection procedure? If non-targeting siRNA was used, please write "mock siRNA" as in figure 5, if not, why was this done?
16. Fig. 4D: It is not clear if the cells used for this experiment are the same as for 4E, also expressing mycBioID-Tollip. The figure legend mentions this, but the main text does not. If the cells do express mycBioID-Tollip, indicate that they were used and why?
17. Fig. 4E: Some of the mutations seem to interfere quite heavily with the Tollip-Parkin-interaction. While R275W-Parkin shows a strong labelling, the T240R variant is labelled very weakly although the lysate contains similar amounts of both variants. Therefore, the conclusion that Parkin-activity is not needed for interaction with Tollip is not fully convincing. Also the role of the UBL domain is still unclear (see point 2. And 3.)
18. Fig. 5A, B: The example pictures do not represent all claimed observations well. For example in the untreated mock siRNA conditions the GFP signal is too strong. In the untreated Parkin siRNA condition the GFP signal is rather weak and diffuse, raising the question if it indeed colocalises with the TOM20 signal. A very similar looking diffuse signal is seen in the AA-treated Tollip siRNA condition, though in this picture it is marked with an arrow indicating no colocalisation of GFP and TOM20. Also in the last condition (AA-treated Parkin siRNA) the TOM20 signal marked with the

lower arrow head is not visible. Please comment.

19. Fig. 5C: The authors state that 9 to 11 cells from 3 independent experiments were counted. This means only 3 or 4 cells were selected from 1 experiment. Correct? How were these cells selected? Given the broad scattering of the results (0 to 50% GFP-positive MDVs in untreated mock siRNA cells) and the apparent low number of events observed (in the 20s? or how can the stepwise appearance of the data points be explained?) this raises concerns how solid this observation really is. Additionally, stating a percent value for MDVs per cell seems counterintuitive. Was the percentage of GFP positive MDVs analysed or the number of GFP-positive MDVs per cell? Please explain or adapt the analysis.

20. Fig. 5E: The colocalisation of TOM20+/CytC- MDVs and Tollip should be quantified. Otherwise, a Parkin-independent role cannot be clearly concluded.

21. Fig. 6B: The statistics appears questionable. TOM1 KO untreated compared to TOM1 AO treated is stated as not significant. It does look like a significant increase though. Is it possible that the line segment was meant to show the comparison of parental untreated with TOM1 KO AO treated? Even if this increase from TOM1 KO untreated to TOM1 AO treated is not significant, this does not mean there is no effect. Tollip clustering could be mediated via another molecule or could be much slower and thereby not observable in the same timeframe. Again, the authors mention 4 - 16 cells each in 3 experiments were analysed, meaning in some value between 12 and 48 cells were analysed. As I understand it, the authors just categorised the cells as Tollip perinuclear cluster containing or not containing. With such a simple categorization system much more cells should be analysed to strengthen the observation. In line with this concern, the figure caption should also be changed (see below).

22. Fig. 6E: The binding of Tollip to LAMP1 is not entirely abolished in the Δ Nterm mutant (in contrast to the CUE mutant). There is a weak colocalisation albeit no clustering. This would fit the previous observation (Fig. 6B) that in the TOM1 KO there is a weak increase in Tollip clustering. This could mean that while TOM1 is important for efficient clustering and LAMP1 association, it is not essential for these processes.

23. Fig. 6 caption: For A and B the authors state "quantification of perinuclear Tollip clusters" and "clusters were quantified ..." This is not reflected in the figure. The authors categorized the cells in two categories and give the percentage of cells in the category "with Tollip perinuclear clusters". This is not a quantification of the clusters.

24. SFig. 2B: The actin band looks overexposed in this example, which would render a quantification problematic. If possible, show a lower exposure of the blot.

25. SFig. 3B, C, D: The quantification shows no change in the Pearson's correlation, though in both overexpression experiments, it looks like Parkin and Pink1 colocalise less with TOM20 in Tollip KO cells. Are there possibly better example pictures?

26. Page 12: Type 'colocalisation'. Correct

Referee #3:

Parkin is a key regulator of damaged induced mitophagy and loss of its function contributes to early onset cases of Parkinson's disease. Emerging evidence is also highlighting a function in the processing of mitochondrial-derived vesicles. Understanding Parkin's function in cells (particularly neuronal cells) will inform why defects in the protein cause early onset PD and hence are important. The current manuscript presents an interesting extension to Parkin's role in MDV formation, identifying that the endosomal protein Tollip as key in trafficking of Parkin-dependent MDVs. The authors present largely compelling and elegant imaging and biochemical data to support an interaction between Tollip and Parkin driven by mitochondrial insult that is important for MDV trafficking to lysosomes.

General comments

1. The manuscript is well-written and clear and the imaging data comprehensive and carefully performed. Notwithstanding that the imaging of MDVs is challenging, I note that some of the quantification is only based on 3-6 cells per independent experiment. Also, it is not clear for example in Figure 2, in these limited number of cells, how many puncta were quantified as the data is expressed as a %. The number of puncta per cell should also be reported in this quantified data. Given the challenges in quantifying the colocalisation of Parkin/Tollip in endosome puncta, the authors support the interaction of Parkin and Tollip using proximity induced tagging (BioID).

Although an informative approach it is not without limitations (see point 3). Hence, testing whether Tollip and Parkin can co-IP (even if only the ectopically expressed tagged variants) would be an important complement to these approaches.

2. The Tollip-dependent decrease in Tom20 in SH-SY5Y cell in 1G is not particularly convincing. As the loss is through MDV trafficking through the endosomal pathway, that the reduction in total Tom20 levels is not marked is not necessarily surprising. Could repeating this experiment with a longer timeframe of AO, but in atg5 ko cells (thereby focusing on the endosomal pathway without interference from autophagosomal pathway) provide a more convincing assessment?

3. BioID is a powerful approach to assess potential protein interactions in the context of intact cells. However, it should be noted that this approach only indicates proximity (<20-30nm), it does not necessarily indicate an interaction and it definitely does not imply a direct or "tight" interaction. Other orthogonal approaches would be required to conclude this. Can Tollip and Parkin be co-IPd? Additionally, the absence of biotinylation by this approach does not necessarily mean that the proteins do not interact. For example, the interacting protein may not have an exposed/available lysine for biotinylation. The authors use "myc-BirA" as the negative control. Presumably this was soluble myc-BirA? Ideally the BirA control should be targeted to the same subcellular compartment (ie endosomes) to be an effective control and to conclude whether the interaction between Tollip and Parkin is specific. Given the data in Figure 2G that Tollip/Parkin puncta markedly increase with AO, would the authors expect an increase in Parkin biotinylation with AO?

The authors conclude that Parkin is still biotinylated in Tom1 ko cells (Figure 3D). However, the biotinylation of Parkin does seem reduced in the data shown. Was this decrease seen in other experiments, which might suggest a role for Tom1 in the Tollip:Parkin proximity/interaction? Can the authors comment on why biotinylation of wild-type HA-Parkin (and also Parkin mutants) was markedly decreased following AO in 4E, but not 3A?

4. Previous reports have shown that R42P in the Ubl domain fails to target mitochondria (Lee et al JCB 2010).

5. The authors conclude that the recruitment of Parkin to Tollip does not require its E3 ligase activity. However, once recruited, Parkin is proposed to ubiquitinate resident proteins in the endosome to mediate further lysosomal trafficking. It would be interesting to understand how Parkin activity is triggered. Do the MDVs contain phospho-S65 ubiquitin for example?

Specific

1. HeLa should be HeLa throughout
2. Page 6: "Further implementing..." "supporting"?
3. Western blot to confirm expression of R78A mutant alongside wt or the CUD domain mutant should be shown in Figure S4D.
4. Page 13, "Surprisingly, Parkin depletion lead..." should read "led".
5. Molecular weight markers should be added to their Western images.

Referee #1

In this study by Ryan et al., the authors identify a role for the Toll-interacting protein Tollip in the targeting of Tom20+ mitochondrial derived vesicles to the endosomal compartment. This occurs through its interaction with Parkin after the vesicles have formed, and appears independent of Parkin's ubiquitination E3 ligase activity, but dependent upon Tollip's ubiquitin binding CUE domain. Loss of Tollip leads to an accumulation of MDVs stimulated with oxidative stress, consistent with a requirement for this protein in later steps of vesicle targeting. The manuscript defines Tollip primarily as an adaptor to sort ubiquitinated receptors through the endocytic compartment, which is of course correct, but it should be noted that this protein is primarily studied as a central adaptor in innate immune signaling (hence the name). Given the rapid emergence of mitochondria as a signaling platform in innate and adaptive immunity, the impact of this study is likely to be much broader than advertised here in terms of quality control. Both the McBride/Desjardins (Cell 2016) and Youle (Nature 2017) labs recently highlighted the role of Pink1 and Parkin as repressors of innate immunity, although they apparently disagree on the mechanisms. I say this not in request of experiments along this line, but to state my enthusiasm for the future implications of this work. The authors should at least comment on the primary function of Tollip in immunity to alert the readers to the majority of the literature on this protein.

Response: We thank the reviewer for their very positive comments on our manuscript and also thank them for pointing out the potential broader impacts of these studies, which positions a key innate immune signalling modulator Tollip in close proximity to mitochondria to regulate mitochondrial quality control. As a result of the reviewer's important points, we have included in the discussion further comments on Tollip's role in innate immune signaling and the potential implications of this based on previously published reports as stated above regarding the connections between mitoQC and innate immunity.

The authors challenge the cells with antimycin/oligomycin (AO) or antimycin alone for a few hours to generate ROS-induced MDVs, and they focus on this in the context of protein quality control. This work differs from previous work analyzing Tom20+ MDVs, which were initially shown to be independent of PINK1/Parkin. In agreement, they show that Tom20 MDV formation does not require Parkin, but that they accumulate. How exactly Tollip and Parkin work together to drive these vesicles to the lysosome is investigated, but the overall conclusions are rather confusing and speculation tends to over-step the data.

Response: Because of pre-existing data to suggest that Parkin is not required for TOM20+ MDV formation, we have performed thorough experiments to evaluate Parkin's role in regulating this subset of MDVs. We believe the data that we include provides substantial evidence to support the idea that Parkin is not substantially required for TOM20 MDV formation, but rather is essential for their trafficking within the endosomal system. For example, in Figure 4C we show that TOM20 MDVs accumulate following Parkin siRNA knockdown. This phenotype was

validated using CRISPR-Cas9 generated Parkin KO cells, shown in Figure EV2. In addition, overexpression of Parkin in HeLa cells reduced the number of TOM20 MDVs (Figure 4H). Importantly, the new data that we have included provides further insight into the potential impact of a Tollip-Parkin association in the trafficking of TOM20 positive MDVs from the mitochondria to the lysosome (Figure 4E-G). We have attempted to clarify some of our discussion and the new additional data that is included provides further validation of this function of Parkin in TOM20 MDV trafficking. Furthermore, we have adjusted our model in Figure 8 to more accurately depict what our data indicates.

Comments and suggestions

- Fig 1G shows Tom20 and COXII protein levels reduced by half in control cells treated with AO for 24 hours, and that only the loss of Tom20 was dependent upon Tollip (Fig 1G). The blots shown are not convincing here, and a 24 hour time point would almost certainly invoke mitophagy and other pathways as the damage accumulates. What do the cells and mitochondria look like at this time point? Have the authors looked at any differences when cells are adapted to galactose rather than in high glucose? In general, it would be helpful if the authors performed this experiment as a time course and probed with a number of additional mitochondrial markers to interrogate mitophagy and potentially protease (LonP) pathways. It is not yet clear what other cargoes are within Tom20 vesicles, so this could be informative regardless of the outcome. As a suggestion, other outer membrane proteins like the mitofusins or Bcl-2 family proteins, cytochrome c for the IMS and PDH should be examined to make their point more clearly. In vitro budding assays from the McBride lab quantified protein release by MDVs with stressors as containing just a few percent of total mitochondrial protein per hour. A loss of 50% of total protein in 24 hours may therefore reflect other mechanisms, including reduced translation, import, etc.

Response: We thank the reviewer for their insight and agree with the reviewers concerns here. Even though the Western blots are not convincing in the original Fig 1G, we trust our quantitative data that illustrates a perturbation in TOM20 degradation following Tollip siRNA. However, after closely reviewing our data, it is very difficult to rule out the contributions of other pathways playing a potential role here. We have attempted additional experiments to address some of these issues by depleting cells of Atg5 by CRISPR-Cas9 prior to Tollip siRNA, in order to determine whether this reduction in TOM20 is mitophagy independent. However, these experiments are complex and therefore the results continue to be difficult to interpret. Therefore, to eliminate ambiguity within these data, we have removed the original Figure 1G-I, as well as the former Supplemental Figure S2 A, B, C, E, and F from the manuscript in order to avoid any confusion. We believe the substantial data that remains in the manuscript fully supports a role for Tollip in TOM20-positive MDV trafficking. With regards to the reviewers question about what types of cargoes are within TOM20 MDVs, this has proven more difficult to determine during the course of our study, but we agree is an important question within the field. This analysis will require more advanced approaches to determine the extent of selectivity for distinct mitochondrial cargo and the composition of these discrete inner/outer membrane

derived vesicles. However, we believe this analysis is beyond the scope of the current manuscript.

- The fluorescent imaging of GFP-Tollip with Parkin, ubiquitin, Tom20, etc., is difficult to see very clearly. There appears to be a lot of background staining throughout the cytosol, with a very high expression in the nucleus in many cases, making it difficult to interpret a few colocalizing dots. This is an issue with the study of MDVs in general, so in each case the authors must take care to validate all antibodies that are used for IF in siRNA or KO cell lines. I appreciate the use of additional cell lines in sup figures, and parkin-GFP, but it remains a concern. Silencing Drp1 would help minimize mitochondrial fragmentation and allow an easier view of MDVs carrying Tom20 or PDH as well, where applicable, particularly when only one of the two markers are being monitored (for localization with Tollip, Parkin, ub, etc).

Response: We agree with the reviewer's comments regarding issues when visualising MDVs and colocalisation more generally. Therefore, in Figure 2 we have amended the immunofluorescence images to eliminate the Hoechst channel, and instead are showing green, red, and blue for the corresponding GFP, Parkin, and ubiquitin channels. This allows for better visualisation of areas of colocalisation.

We understand the reviewers concerns that it is important to validate antibodies and ensure the discrete localisation we are showing is physiological. An important point to consider is that Tollip is a multifunctional protein, so localises to a number of intracellular compartments as well as to the plasma membrane. We have been careful to validate our localisation data across different cell lines as well as using both antibodies and tagged versions of both Tollip and Parkin. Importantly, it is challenging to validate their localisation to MDVs due to their low numbers and small size, which additionally requires triple labelling procedures as well. But we have included substantial supporting data that validates across multiple platforms using different approaches to provide strong evidence of a physiological role. We therefore do not believe further validation using Drp1 silencing is necessary in this case.

- Do the authors see changes in PDH+ MDVs upon loss of Parkin, as previously observed? There is very little consideration of the potential differences between Tom20 and PDH MDVs and why they would transit to the late endosome in different pathways. Do they think that Tom20 MDVs target Rab5/EEA1 early endosomes and PDH directly targets multivesicular bodies?

Response: We agree this is an important point made by the reviewer, so we have quantified PDH+ MDVs from SH-SY5Y cells treated with Parkin siRNA. The results indicate that loss of Parkin does not lead to an increase in PDH+ MDVs as seen for Tom20+ MDVs (Figure 4C and 4D). Instead there is a subtle, but significant decrease in steady state PDH MDVs as published by McLelland et al., EMBO J, 2014. We have also done the same analysis for Tom20 and PDH positive MDVs in a CRISPR-Cas9 generated Parkin KO SH-SY5Y cell line, which corroborates the

siRNA data (Figure EV2). Therefore, the data to support the differential effects of Parkin loss of function on both TOM20 and PDH MDVs has been confirmed by two independent methods, which provides significant support for these phenotypes. To address the reviewers 2nd point, we believe due to the unique membrane topology that is characteristic of TOM20 and PDH MDVs there are likely unique mechanisms that are required to interact with the endosomal pathway. We have evidence to suggest TOM20 MDVs are trafficked via an early endosome, Figure 5A-D. As the reviewer suggests, it may be the case that PDH MDVs require direct trafficking to an MVB due to their double membrane structure. This is a question that is important for the field to answer, but we feel is out of the scope of this manuscript.

- GFP-Tollip localizes to cytosol in untreated cells, but shows a very strong translocation to the nucleus upon treatment (see Fig 2, particularly with the mutants). Is this a real translocation, or does it reflect cleavage of the GFP during the stress?

Response: Although this may be interesting, after review of our images, we believe this is a variable effect and therefore have no evidence to suggest this is a physiological localisation. We amplify the GFP signal using a rabbit polyclonal antibody, so this may be a source of some of the nuclear non-specific staining. In addition, due to variable expression levels, some of this may be due to breakdown and cleavage of the GFP as mentioned by the reviewer.

- The data is often presented as % of Tollip (or ubiquitin, etc) puncta that are positive for Y, but how many puncta are there? How is a puncta defined? Tollip is known to act within the endosomal compartments, so how many of these "puncta" are endosomes? Some of this is shown in S6 and S7, but quantification is as a pearsons co-efficient, not as "puncta". So trying to reconcile the presentation of data between different figures is difficult.

Response: We have included new information within the Figure 2 figure legend providing details of the range of puncta that were counted per cell. We counted all puncta within the cell, with the assumption all was membrane associated. Due to Tollip's diverse subcellular localisation, it would not be possible to divide this further between individual endosome subtypes.

We have performed Pearson's correlations shown in Appendix Figure S2A to validate the % puncta data in Figure 2C. This analysis was done to validate our counting method, which illustrates an unbiased method performs similarly to our manual counts.

- Loss of the lipid binding domain of Tollip increased the colocalization with Parkin, whereas loss of the ubiquitin binding CUE domain abolished this, both in fluorescence and with BioID approaches with Tollip as bait. This indicates Tollip requires ubiquitin to interact with Parkin. However, mutations in the Parkin ub ligase domain actually increased the interaction with Tollip even without AO treatment, which is lost upon treatment (Fig 4E R275W mutant in BioID). This result is not well described in the results section and is very confusing. Where are they interacting? What

happens to PDH+ MDVs in these conditions? Where is the CUE requirement coming from, if parkin ub mutants bind so much more strongly? Does the CUE mutant bind the R275E mutant?

Response: We believe that the interaction of Tollip with Parkin is occurring independently of its role in trafficking of MDVs through the endolysosomal system. Therefore, although Tollip interacts with a ligase dead mutant of Parkin, this does not rule out the possibility that Tollip may be required for Parkin function to regulate MDV trafficking. This is supported by the results which show the lipid binding mutant increases its association with Parkin on endosomes, while the CUE mutant abolishes its association. The CUE domain of Tollip may be interacting with endosomal or mitochondrial membrane cargo. However, it is important to note these results don't necessarily indicate ubiquitylation by Parkin is required, but instead show a requirement for Tollip-ubiquitin interactions. We believe our results suggest that Tollip acts as a molecular switch between cargo and lipid binding to regulate Parkin function. We also don't have evidence this is via a direct association, as there may be an intermediate required that facilitates Tollip-Parkin interactions. In relation to this, there could be similar mechanisms at play as described for Tollip's role in endosomal positioning (Jongsma ML, Cell, 2016). We now have included further data using Parkin with a deleted UBL domain, which indicates that loss of this domain reduces its interaction with Tollip (Figure 4I), which give further insight into the Tollip-Parkin association. We do agree with the reviewers there may be both ubiquitin-dependent and -independent mechanisms at play here, and therefore, we have softened our wording in the discussion to account for this.

- The authors show that loss of Parkin leads to an increase in Tom20 MDVs, consistent with previous work showing Parkin is not required for the formation of this class of vesicles. The distinction in the current study is that they observe some recruitment of Parkin to Tom20 MDVs after they form, something not observed in previous work. Unlike Tollip, the increased number of Tom20 MDVs is not due to a requirement for Parkin to target and enter the endocytic compartment, since the localization with the FYVE probe was retained. Again, this is confusing. Perhaps a likely explanation is that loss of Parkin results in increased mitochondrial damage, which increased Tom20 MDV generation and flux, without a real "block" anywhere. But in terms of the Parkin recruitment to these MDVs, if Parkin is required to recruit Tollip to the MDVs, and Tollip is required to deliver the MDVs to the endosome, why isn't Parkin required for that step as well? The authors suggest Parkin is requisite for the latter step of fusion/maturation into a late endosome, but this is not borne out from the data presented.

Response: To address the reviewers concern that loss of Parkin is instead inducing mitochondrial damage and therefore MDV generation rather than blocking MDV trafficking, we have employed experiments to evaluate mitochondrial function and trafficking of MDVs to a LAMP1 compartment in Parkin KO cells. Importantly, these CRISPR-Cas9 generated Parkin KO cells exhibit an accumulation of TOM20 MDVs (Figure EV2), similar to our siRNA experiments in Figure 4C and D. We have additionally measured ATP production in Parkin KO cells cultured in

the presence of galactose containing culture media. These new results in Figure EV3 indicate that loss of Parkin or Tollip has no effect on mitochondrial ATP production, which is specifically blocked upon inclusion of oligomycin. In addition, the mitochondrial network is intact compared to wild-type as shown in our siRNA data (Figure 4A) as well as in the KO cells (Figure EV2A), which provide further support for a lack of increased mitochondrial damage following Parkin loss of function. Furthermore, we have evaluated the trafficking of TOM20 MDVs to the lysosome in Parkin KO cells, by determining their extent of colocalisation with LAMP1. These new data in Figure 4E-G indicates that Parkin loss of function leads to a reduction in TOM20 MDV colocalisation with LAMP1, suggesting a lysosomal trafficking defect, similar to the results identified following Tollip loss of function (Figure 7A and B). We believe our current data supports our model that Tollip may be required for MDV interactions with the endosomal network, and that Parkin additionally functions on these compartments to facilitate trafficking onward through the endosomal system. Our data also indicates that neither Parkin nor Tollip are required for TOM20 MDV formation at steady state or following stress conditions.

- When examining Tollip localization with LAMP1 the authors state "we observed an increase in the trafficking of Tollip to LAMP1 compartments (Figure 6F)". Do they mean Tollip arrived there on MDVs, or was recruited from cytosol to the late endosome? The signals for recruitment are distinct, and I am not clear what is meant by this, or whether they can interpret the colocalization as being due to traffic. There is generally a lot of speculation in the discussion about the mechanisms and steps between MDV, early and late endosome. They speculate that MDVs "piggy-back" on Tollip endosomes, which is confusing given their data showing apparent complete colocalization (as a single vesicle).

Response: We think it is very unlikely that Tollip is trafficked to LAMP1 via MDVs alone as MDV-facilitated trafficking only would likely not be sufficient to deliver the observed amounts of Tollip to the lysosome. Due to Tollip's known role as an endosomal adaptor, it is more likely Tollip is also trafficked via endosomes or possibly recruited from the cytosol. Our data in Appendix Figure S4 suggests Tollip is at least partially trafficked to a late endosome/lysosome in response to mitochondrial damage via an early Rab5 positive compartment.

Importantly, our data illustrates that MDVs colocalise with EEA1, Rab7a, Tollip and LAMP1, as well as gaining FYVE-positive membranes. This would suggest an endosomal pathway being required for their trafficking. We would suggest that MDVs may fuse with an endosomal compartment or acquire endosomal membrane, which is dependent on Tollip's role as an endosomal adaptor protein. Tollip's role as an endosomal adaptor could either provide a switch to ensure correct cargo sorting and trafficking to a lysosomal compartment or be required for endosomal positioning to ensure fusion events occur along the endosomal route.

Overall, the primary conclusions are that Tom20 MDVs, but not PDH MDVs, require Tollip to transport and fuse with the endocytic compartment. Parkin interacts with Tollip on a subset of

Tom20 MDVs, and its loss also increases the number of these MDVs. However, the details of how these two proteins work together, and at which level of endosome, is not well resolved at this point. The strength of the study is the novel association of this adaptor, so well studied within the immune system, with the Parkin regulated pathways of mitochondrial vesicle formation.

Response: We appreciate the reviewers concerns that we haven't fully elucidated the mechanisms of a Tollip-Parkin complex to regulate MDV trafficking through the endocytic pathway. However, our data provides substantial evidence this interaction may be required for MDVs to enter the endocytic pathway, likely via an early stage compartment, and facilitate the transition to a late endocytic compartment in order to become competent for lysosome fusion. As the reviewer states, we have provided abundant data to support a tight relationship between Tollip and Parkin to regulate Tom20+ MDV trafficking to the lysosome. This is a significant advance in the field, which suggests Parkin may be functioning both at the mitochondria and at the endosome to regulate trafficking of damaged mitochondrial cargo. This study additionally provides substantial evidence of the endosomal adaptor Tollip playing a key role in this process. As stated by the reviewer, the findings from this study will also be significant to those interested in determining the relationship of mitochondrial damage to innate immune signalling with potential contributions to our understanding of the pathophysiological mechanisms of conditions such as Parkinson's.

Referee #2

In this manuscript the authors investigate the role of Tollip in a particular pathway linked to mitochondrial stress response and quality control. This pathway involves the formation of MDVs and its trafficking to lysosomes via the endosomal pathway and is distinct to the classical Parkin-dependent mitophagy pathway. The authors nicely address different stages and dependencies along this route. Despite the interaction of Tollip with Parkin, Tollip does not participate in the classical mitophagy pathway, but instead has an important role in the MDV trafficking pathway. The authors show that Tollip interacts with a subset of MDVs and helps in their delivery to an endosomal compartment of the cell.

In particular, the authors show that knockdown of Tollip increases the occurrence of TOM20 positive MDVs and that Tollip colocalises to some of these MDVs. They show a stress-induced colocalisation of Tollip with Parkin and a dependency of this interaction on the ubiquitin-interacting CUE domain of Tollip. This Parkin-Tollip interaction was confirmed by proximity labelling. Some of the Tom20-positive MDVs colocalised with 2xFYVE in a Tollip-dependent but Parkin-independent manner. Tollip clustering occurred at LAMP1 positive sites in the cell. This was shown to be partially dependent on the presence of TOM1 and Tollip's interaction domain with TOM1. Finally, the authors showed that some of these MDVs are also positive for LAMP1 and that this colocalisation is dependent on Tollip.

Overall, this is a very comprehensive and well-performed study given important insights into this novel pathway of mitochondrial QC. It focuses on the role of Tollip but also gives novel insights

into the role Parkin in MDV trafficking and targeting to lysosomes via endosomes. A few issues still need to be addressed or clarified in my opinion as they are not clear or not convincingly shown.

Response: We thank the reviewers for their positive comments on the manuscript and support of the novelty of this study.

Major issues:

1. The accumulation of Tom20 MDVs upon depletion of Tollip is taken as an indication of Tollip playing a role in trafficking rather than playing a role in its formation. Although this is quite likely, still, the authors did not exclude that Tollip induces some sort of mitochondrial dysfunction that could initiate MDV formation. The authors should test whether mitochondrial respiration or morphology is altered upon depletion of Tollip or explain why they think they can exclude this.

Response: This is a valid concern made by the reviewer, but our collective data has aimed to resolve this issue. In particular, our immunofluorescence microscopy in Figure 1B indicates that following loss of Tollip there is no significant perturbation of the mitochondrial network. Additionally, overexpression of GFP-Tollip (Figure 7C) did not appear to alter the morphology of the mitochondrial network. To address the reviewer's concerns that the mitochondria may be damaged or dysfunctional following Tollip loss of function, we measured ATP production following culture in galactose-containing media in Tollip KO SH-SY5Y cells generated by CRISPR-Cas9 approaches. These new data in Figure EV3A and B indicate similar levels of ATP production in the presence of galactose media, which is significantly decreased in the presence of oligomycin, while still intact in Glucose + oligomycin containing media. In addition, we observe no change in total mitochondrial load as indicated by protein expression of resident mitochondrial proteins (Figure EV3C). Therefore, our collective data supports the idea that Tollip loss of function is impacting on TOM20 MDV trafficking rather than triggering mitochondrial damage. This is additionally supported by the selective nature of the cargo trafficking, as PDH MDVs are not perturbed following Tollip loss (Figure 1D).

2. Regarding the text describing figure 4 (page 9 paragraph 2): Have the authors considered that the Tollip-Parkin interaction might be mediated directly via the UBL domain of Parkin instead of ubiquitin itself? Is CUE-domain-dependent binding of Tollip occurring directly to the UBL or indirectly via ubiquitylated Parkin? This would be an important point to address or to discuss more clearly in my opinion.

Response: We considered the UBL domain may be important for the Tollip interaction based on previously published reports identifying the mechanisms of Eps15 interactions with the Parkin UBL to regulate endocytic trafficking (Fallon et al., Nat Cell Biol, 2006). Although from our initial data the R42P mutant still retained an interaction with Tollip (new Figure EV4B), we have performed additional experiments to address the reviewer's concerns. We now have included

new data which indicates that Parkin lacking the N-terminal UBL domain ($\Delta 1-76$ aa) in our BioID-Tollip assay significantly disrupts their interaction (Figure 4I). This is consistent with Fallon et al., which showed that while Eps15 requires the UBL to interact with Parkin, the R42P mutation does not disrupt their association in mammalian cells. We don't as of yet know whether the Tollip-Parkin interaction is direct or mediated by a secondary intermediate. This is an area that we are actively interested in, but believe this warrants continued investigation and is beyond the scope of this manuscript.

3. Along this line, how does the used R42P Parkin mutant affect MDV formation under mitochondrial stress (see Fig. 4F)? Does it also impair MDV formation?

Response: We have performed some preliminary experiments in HeLa cells to indicate that expression of the R42P has no effect on TOM20 MDV formation. These data are indicated below. This does not rule out the possibility that this mutant may affect the trafficking of these MDVs to the lysosome. But this is an area for future investigation, which we believe is beyond the focus of this manuscript. Previous studies have already established the impact of R42P Parkin on inner membrane derived MDV formation (McLelland et al., EMBO J, 2014), which indicates that R42P still localises to this subtype of MDVs and there is no significant effect on the formation of these Parkin positive MDVs. These results are consistent with our results with respect to TOM20 MDVs.

4. Fig. 5G: Showing the standard error of the mean (SEM) in a n=2 experiment is questionable. It's better to show the standard deviation (SD) which is equal to the range of the two experiments, or show the two values as dots and the average as a line. How can the authors do statistics on n=2 experiments? Please adapt or include more experiments.

Response: We apologise to the reviewer for this oversight, we have therefore performed an additional experiment and adjusted the data in Figure 5G, which now represents 3 independent experiments.

5. In the discussion (last paragraph) the authors might also discuss that under stress most Parkin is rapidly located at mitochondria and thus Tollip-Parkin interactions might occur here as well, not only via binding to cytosolic Parkin.

Response: Our data which uses Parkin mutants that lack mitochondrial localisation, T240R and R275W, indicates a retention of a Tollip interaction suggesting interactions may be occurring on other compartments or within the cytosol. However, we agree that we cannot rule out interactions occurring at the mitochondria. Therefore, as suggested by the reviewers we have adjusted our wording in the discussion to account for this possibility.

Minor issues:

6. The significance asterisks should be used uniformly throughout the manuscript. In figure 2 ** is given as $p < 0.005$ while the same p-value for figures 4, 6 and 7 is represented by ***

Response: We apologise for this error – we have corrected these and are now consistent throughout.

7. It is not entirely clear whether the quantification analyses are different for Fig 2C, G, H, I, J and 5C. The authors should explain this in more detail. See also the Comment to 5C for this. If they indeed quantified the colocalisation differently, why did they do so?

Response: This is a valid point made by the reviewer, therefore we have attempted to better clarify the analyses that have been performed in relation to the figure panels indicated. These details are now included within the figure legend.

8. Fig. 1E: A quantification for the occurrence of these events would be helpful.

Response: We have done this quantification on a limited dataset, which is illustrated below. However, we do not feel this warrants further quantitation and inclusion in the manuscript – as these qualitative data presented in the manuscript only aim to determine whether Tollip does in fact localise to TOM20 MDVs, which we indicate in 2 independent cell lines, SH-SY5Y cells (Figure 1E) as well as in Parkin expressing Hela cells (Figure EV1C).

9. Fig. 1B, G: Was non-targeting siRNA used as negative ctrl (mock). What does 'mock' mean? Please clarify.

Response: The mock control lacks siRNA – instead this represents transfection reagent only. We have adjusted all annotations on the figures to read 'mock' and have made the explanations clearer in the methods section.

10. Fig. 1G, H, I: This conclusion is not convincing. It is also unlikely that the pool of Tom20 is largely affected given the fact that most of Tom20 remains on intact mitochondria and only a minor fraction is at MDVs anyway. Moreover, the example blot does not quite reflect the quantification, is there a better example? Some bands look very strong. Are the authors sure that there is no saturation, which would affect the quantification? I feel that this is overstated, but also not required. Please correct or explain.

Response: We have considered the reviewers comments and agree with reviewer 1 and 2 that although our data supports a defect in Tom20 degradation in Tollip loss-of-function cells, there may be multiple mechanisms working here and so it is impossible to deduce exactly what level of degradation is occurring via MDVs. We have attempted experiments to perform Tollip siRNA in the context of Atg5 KO, as per Reviewer 3 comments, to address some of these questions but the results of these experiments in SH-SY5Y cells have proven difficult to interpret and continue to provide ambiguity. We have therefore removed these data, in Figure 1 and its associated data in the previous Supplementary figures, from the manuscript to avoid confusion and ambiguity.

This is an area that we will investigate in the future to determine whether Tollip functions across multiple routes of mitochondrial quality control. However, we believe the substantial data we have included within the manuscript supports a role for Tollip alongside Parkin in MDV trafficking.

11. Fig. 2A: How many of these events were observed? Can the authors give an estimate or a quantification for the occurrence of these events?

Response: These events are very difficult to quantify, from our estimates there are around 5% of Tom20 MDVs that are Parkin positive. Also, importantly, we can only capture these moments when we deplete cells of Tollip, which perturbs the trafficking of the MDVs to the lysosome. We believe this is an important effect to note and gives support of our hypothesis that in the case of Tom20 MDVs, Parkin has a transient association that regulates mitochondrial cargo trafficking through the endocytic system. In Figure 2A we are also identifying endogenous Parkin in SH-SY5Y cells which can be challenging to evaluate, which is likely due to its low level abundance and limited quantity on these small vesicles.

12. Fig. 2B, E, F: it is not clear which colour was attributed to ubiquitin in the merge picture. Was white chosen as a color? If so, this is not ideal to visualizing a colocalisation. The authors should rather use a different false colouring for ubiquitin.

Response: We apologise for the confusion. We have amended this figure and now only show three colours – indicated by Tollip in green, Parkin in red, and ubiquitin in blue. This should address the reviewer's concerns and allow for better visualisation of the colocalisation.

13. Fig. 3: Although highly unlikely, there is the possibility that Tollip somehow interacts with the HA-tag instead of Parkin. Technically, a proper control e.g. HA-GFP to exclude this is missing.

Response: We understand the reviewers concerns, but we believe this is an unlikely occurrence based on the use of a small HA tag. In addition, the data with the CUE domain mutant of Tollip gives support that non-specific interactions with the HA tag is not occurring, thus providing a good control for the specificity of this interaction. In addition, our new data in Figure 4I illustrates no biotinylation of HA-tagged Δ UBL Parkin by BioID-Tollip, thus giving further support for the specificity of Tollip-Parkin associations.

14. Fig. 3 caption. There is a typo in the figure caption. "Tollip specifically interacts with Tollip". Please correct.

Response: Thank you for bringing this to our attention, this has been corrected in the Fig. 3 caption.

15. Fig. 4A, B, C: Was non-targeting siRNA used as negative ctrl (mock) or does mock mean no siRNA at all, just transfection procedure? If non-targeting siRNA was used, please write "mock siRNA" as in figure 5, if not, why was this done?

Response: For all experiments the mock group represents transfection reagent only, without the non-targeting siRNA. We have amended all annotations to 'mock' and have also made the description of this clearer in the methods section.

16. Fig. 4D: It is not clear if the cells used for this experiment are the same as for 4E, also expressing mycBioID-Tollip. The figure legend mentions this, but the main text does not. If the cells do express mycBioID-Tollip, indicate that they were used and why?

Response: The cells in what is now Fig. EV4A are stable cell lines that only express the HA-tagged wild-type and mutant versions of Parkin. These experiments were performed to validate our mutants and illustrate differences in the mitochondrial targeting capacity in response to mitochondrial depolarisation. We apologise, the figure legend was incorrect and therefore we have amended this appropriately to reflect these cells in EV4A are only expressing HA-tagged Parkin.

17. Fig. 4E: Some of the mutations seem to interfere quite heavily with the Tollip-Parkin-interaction. While R275W-Parkin shows a strong labelling, the T240R variant is labelled very weakly although the lysate contains similar amounts of both variants. Therefore, the conclusion that Parkin-activity is not needed for interaction with Tollip is not fully convincing. Also, the role of the UBL domain is still unclear (see point 2. And 3.)

Response: It is very difficult to quantify the results from the BioID assays, as the labelling period is 24 hours in this case, and there is inherent variability across experiments in the levels of labelling. Therefore, we only use these data to illustrate that none of the Parkin mutants tested disrupt the interaction with Tollip. We don't feel comfortable with making strong conclusions from the levels of labelling across the different mutants. We do find it interesting that the R275W Parkin interacts strongly with Tollip at steady state, which gives support for either a cytosolic interaction or Tollip functioning to recruit Parkin onto a subset of endosomes. In addition, this mutant form of Parkin may lead to an increased dwell time on endosomes, due to disrupted E3 ligase activity, thus allowing for this elevated biotinylation.

We have further addressed the reviewer's comments regarding the UBL mutant (R42P) and have performed additional experiments using Parkin with a UBL deletion (Δ 1-76 aa). The results are shown in Figure 4I, which indicates there is a significant reduction in a Tollip-Parkin interaction. These data are consistent with the mechanism of the Eps15-Parkin association (Fallon, L. et al.,

Nat Cell Biol, 2006), which indicates although loss of the UBL is required for Eps15-Parkin interactions, the R42P mutation does not disrupt their association in mammalian cells.

18. Fig. 5A, B: The example pictures do not represent all claimed observations well. For example, in the untreated mock siRNA conditions the GFP signal is too strong. In the untreated Parkin siRNA condition the GFP signal is rather weak and diffuse, raising the question if it indeed colocalises with the TOM20 signal. A very similar looking diffuse signal is seen in the AA-treated Tollip siRNA condition, though in this picture it is marked with an arrow indicating no colocalisation of GFP and TOM20. Also, in the last condition (AA-treated Parkin siRNA) the TOM20 signal marked with the lower arrow head is not visible. Please comment.

Response: We understand the reviewers concerns regarding the image intensity of the GFP-FYVE construct, but it is important to note we have carefully evaluated these results across multiple cells within 3 independent data sets. There is variability in expression levels, since these are transient transfections, but this variability is appropriately accounted for in our quantitative results. However, we take the reviewers concerns onboard and have adjusted the levels in some our images to be more consistent across the different groups, to better illustrate the representative result. For Figure 5A, we have adjusted the GFP signal in the mock untreated group to address the reviewers concern that the GFP signal is too strong. For Figure 5B, in the AA treated group, we have adjusted the TOM20 signal in the Parkin siRNA group so each of these spots is now clearly visible.

In order to further address the reviewers concerns that the quantitative data is not represented well by the images included, we have performed further quantitation and have increased our cell counts across all groups. This has provided more statistical power to our results and continues to show significant trends, despite some variability. These are challenging experiments due to the transient expression of the GFP-FYVE probe and its variable expression in SH-SY5Y cells, but we have ensured our quantitation was performed on cells with comparable levels of GFP expression.

19. Fig. 5C: The authors state that 9 to 11 cells from 3 independent experiments were counted. This means only 3 or 4 cells were selected from 1 experiment. Correct? How were these cells selected? Given the broad scattering of the results (0 to 50% GFP-positive MDVs in untreated mock siRNA cells) and the apparent low number of events observed (in the 20s? or how can the stepwise appearance of the data points be explained?) this raises concerns how solid this observation really is. Additionally, stating a percent value for MDVs per cell seems counterintuitive. Was the percentage of GFP positive MDVs analysed or the number of GFP-positive MDVs per cell? Please explain or adapt the analysis.

Results: The data are represented as the % of total MDVs positive for GFP-FYVE per cell. The number of GFP-positive MDVs were counted and represented as a percentage of the whole

population of MDVs, this has been made clearer on the figure axes. We represented the data this way to most appropriately illustrate changes in endosomal trafficking of the MDVs across each experimental group.

We do agree with the reviewer, there is a level of variability and we understand the concerns with the low cell counts. We apologise for these misleading results based on a low N. We have therefore increased the number of cells counted/experiment, which has given the data more statistical power. The data now represents approximately 7-9 cells/experiment, from 3 independent experiments.

20. Fig. 5E: The colocalisation of TOM20+/CytC- MDVs and Tollip should be quantified. Otherwise, a Parkin-independent role cannot be clearly concluded.

This is a challenging experiment to perform due to the likely transient nature of the association of Tollip on MDVs and the limited number of Tollip-positive MDVs at any one given time. We therefore only include these qualitative results to illustrate that in the absence of Parkin there appears to be little to no effect on Tollip localisation. We have additionally performed experiments in SH-SY5Y Parkin KO cells generated by Lentiviral CRISPR-Cas9, and again these results indicate no apparent perturbation in Tollip association with MDVs. However, due to the difficulty in quantifying this localisation with any degree of confidence, and as a result the lack of definitive evidence ruling out a Parkin-dependent role, we have toned down our wording in the text as per the reviewer's concerns.

21. Fig. 6B: The statistics appears questionable. TOM1 KO untreated compared to TOM1 AO treated is stated as not significant. It does look like a significant increase though. Is it possible that the line segment was meant to show the comparison of parental untreated with TOM1 KO AO treated? Even if this increase from TOM1 KO untreated to TOM1 AO treated is not significant, this does not mean there is no effect. Tollip clustering could be mediated via another molecule or could be much slower and thereby not observable in the same timeframe. Again, the authors mention 4 - 16 cells each in 3 experiments were analysed, meaning in some value between 12 and 48 cells were analysed. As I understand it, the authors just categorised the cells as Tollip perinuclear cluster containing or not containing. With such a simple categorization system much more cells should be analysed to strengthen the observation. In line with this concern, the figure caption should also be changed (see below).

Response: This is a good point brought up by the reviewer and agree further quantitation was needed to verify our phenotypes. To address the reviewer's concerns, we have performed further quantitation on 2 additional experiments, counting more than 50 cells/experiment and have included these data within Figure 6B. Despite the additional experiments, the results continue to show a similar trend in the data. Therefore, we agree with the reviewer's interpretation and have adjusted our wording to consider this result only indicates a

suppression, rather than a complete loss of Tollip translocation. In addition, this residual localisation of Tollip could be due to low level expression of related Tom1 family members, such as Tom1L2 that may be present within HEK293 cells. Tom1L2 is highly related and therefore able to compensate for Tom1 function (Tumbarello et al., Nat Cell Biol, 2012) and may be partially responsible for this phenotype.

22. Fig. 6E: The binding of Tollip to LAMP1 is not entirely abolished in the Δ Nterm mutant (in contrast to the CUE mutant). There is a weak colocalisation albeit no clustering. This would fit the previous observation (Fig. 6B) that in the TOM1 KO there is a weak increase in Tollip clustering. This could mean that while TOM1 is important for efficient clustering and LAMP1 association, it is not essential for these processes.

Response: We agree with the reviewer's observation there is some level of localisation of the Δ Nterm mutant to a LAMP1 compartment, which is consistent with the results in Figure 6B. Based on these data we have softened the wording to indicate that while Tom1 is important for efficient Tollip translocation to a LAMP1 compartment, it is not essential. Although, it should be noted our data from Hela Tom1 KO cells in Figure 6G does indicate a complete loss of AO-induced Tollip/LAMP1 colocalisation. Furthermore, as stated above, some of these differences are likely a result of compensation by related Tom1 family members, such as Tom1L2, which may have context and cell-type differences.

23. Fig. 6 caption: For A and B the authors state "quantification of perinuclear Tollip clusters" and "clusters were quantified ..." This is not reflected in the figure. The authors categorized the cells in two categories and give the percentage of cells in the category "with Tollip perinuclear clusters". This is not a quantification of the clusters.

Response: We apologise for this confusion, we have clarified our descriptions in the Figure legend and have adjusted the y-axis labels accordingly.

24. SFig. 2B: The actin band looks overexposed in this example, which would render a quantification problematic. If possible, show a lower exposure of the blot.

Response: All of the western blotting in the manuscript was done on a LiCOR Odyssey Infrared imaging system. This ensures there is no image saturation and provides the ability to quantitate differences due to the dynamic linear range. To the specific point made by the reviewer, these data being identified in the original Supplemental Fig S2B have been removed from the manuscript for reasons as stated earlier in our rebuttal (Point 10).

25. SFig. 3B, C, D: The quantification shows no change in the Pearson's correlation, though in both

overexpression experiments, it looks like Parkin and Pink1 colocalise less with TOM20 in Tollip KO cells. Are there possibly better example pictures?

Response: We thank the reviewers for highlighting this discrepancy in our data. Although there is cell to cell variability in PINK1 and Parkin expression levels as well as translocation, our quantitative data shows no difference between wild-type and Tollip KO cells. To ensure our images are representative of our quantitative data, we have included new images for PINK1 and Parkin in the Tollip KO cells which illustrates similar translocation of both Parkin and PINK1 to TOM20 structures compared to wild-type (new Appendix Figure S1B and C).

26. Page 12: Type 'colocalisation'. Correct

Response: Thank you for bringing this to our attention, this has been corrected.

Referee #3

Parkin is a key regulator of damaged induced mitophagy and loss of its function contributes to early onset cases of Parkinson's disease. Emerging evidence is also highlighting a function in the processing of mitochondrial-derived vesicles. Understanding Parkin's function in cells (particularly neuronal cells) will inform why defects in the protein cause early onset PD and hence are important. The current manuscript presents an interesting extension to Parkin's role in MDV formation, identifying that the endosomal protein Tollip as key in trafficking of Parkin-dependent MDVs. The authors present largely compelling and elegant imaging and biochemical data to support an interaction between Tollip and Parkin driven by mitochondrial insult that is important for MDV trafficking to lysosomes.

Response: We thank the reviewer for their positive comments on the impact of the study and the quality of the data within the manuscript.

General comments

1. The manuscript is well-written and clear and the imaging data comprehensive and carefully performed. Notwithstanding that the imaging of MDVs is challenging, I note that some of the quantification is only based on 3-6 cells per independent experiment. Also, it is not clear for example in Figure 2, in these limited number of cells, how many puncta were quantified as the data is expressed as a %. The number of puncta per cell should also be reported in this quantified data. Given the challenges in quantifying the colocalisation of Parkin/Tollip in endosome puncta, the authors support the interaction of Parkin and Tollip using proximity induced tagging (BioID). Although an informative approach it is not without limitations (see point 3). Hence, testing whether Tollip and Parkin can co-IP (even if only the ectopically expressed tagged variants) would be an important complement to these approaches.

Response: We thank the reviewers for their positive comments on the manuscript and the comprehensive data we have included. In addition, we agree with the reviewers concerns regarding some of the quantitative data, therefore we have increased the number of cells counted/experiment for a number of experiments, including data in Figure 4C, 5C, and 6B. In addition, we have included details in the Figure 2 legend, which describes the number of puncta per cell that were quantitated for these data.

Addressing the reviewers concerns regarding the BioID method to illustrate a Tollip-Parkin interaction has been quite the challenge. We think much of this is due to a very weak, transient association that may be occurring via indirect methods. However, we have been successful to illustrate, although weakly, that overexpressed HA-tagged Parkin can coimmunoprecipitate GFP-Tollip following AO-induced mitochondrial damage. This is now Figure 3C. We hope this is sufficient to address the reviewer's concerns. In addition, we believe the data using the CUE mutant of Tollip provides a valuable control showing this association with Parkin is physiological and specific. Furthermore, we have new data which suggests the UBL domain of Parkin is required for this association with Tollip (Figure 4I), which further supports the specific nature of this interaction.

2. The Tollip-dependent decrease in Tom20 in SH-SY5Y cell in 1G is not particularly convincing. As the loss is through MDV trafficking through the endosomal pathway, that the reduction in total Tom20 levels is not marked is not necessarily surprising. Could repeating this experiment with a longer timeframe of AO, but in atg5 ko cells (thereby focusing on the endosomal pathway without interference from autophagosomal pathway) provide a more convincing assessment?

Response: We have considered the reviewers comments and agree with all reviewers that although our data supports a defect in Tom20 degradation in Tollip loss of function cells, there may be multiple mechanisms working here and so it is impossible to deduce exactly what level of degradation is occurring via MDVs. We have produced Lentiviral CRISPR generated Atg5 KO SH-SY5Y cells and performed Tollip siRNA on these cells, as per the reviewer's comments above, to address some of these questions but the results of these experiments have proven difficult to interpret and continue to provide ambiguity. We have therefore removed these data in Figure 1G-I and corresponding data in the original Supplemental Figure S2 from the manuscript to avoid confusion and ambiguity. This is an area that we will investigate in the future to determine whether Tollip functions across multiple routes of mitochondrial quality control. However, we believe we have included substantial and thorough data within the manuscript to support a role for Tollip alongside Parkin in MDV trafficking.

3. BioID is a powerful approach to assess potential protein interactions in the context of intact cells. However, it should be noted that this approach only indicates proximity (<20-30nm), it does not necessarily indicate an interaction and it definitely does not imply a direct or "tight"

interaction. Other orthogonal approaches would be required to conclude this. Can Tollip and Parkin be co-IPd?

Response: As mentioned in point 1, we have included a co-IP experiment using cell lines stably expressing ectopic HA-Parkin and GFP Tollip, which shows a weak association of Tollip and Parkin following AO-induced mitochondrial damage. These data were challenging to obtain and we believe collectively the interaction data suggests the association of Parkin and Tollip may be weak and transient, likely via indirect/secondary mechanisms which we are currently investigating. However, we believe delineating these mechanisms is out of the scope of this initial publication.

Additionally, the absence of biotinylation by this approach does not necessarily mean that the proteins do not interact. For example, the interacting protein may not have an exposed/available lysine for biotinylation. The authors use "myc-BioID" as the negative control. Presumably this was soluble myc-BirA? Ideally the BirA control should be targeted to the same subcellular compartment (ie endosomes) to be an effective control and to conclude whether the interaction between Tollip and Parkin is specific.

Response: We understand the reviewers concerns regarding specificity and proper controls for these experiments. But we believe that the CUE mutant of Tollip provides the best control here, which illustrates a loss of the Parkin interaction, while maintaining interactions with Tom1. This is a valuable piece of data, which supports the notion this interaction is physiological and specific. In addition, we have included new data which illustrates loss of an interaction between Tollip and Parkin lacking its UBL domain (Figure 4I), which provides further validation these interactions are physiological.

Given the data in Figure 2G that Tollip/Parkin puncta markedly increase with AO, would the authors expect an increase in Parkin biotinylation with AO?

The authors conclude that Parkin is still biotinylated in Tom1 ko cells (Figure 3D). However, the biotinylation of Parkin does seem reduced in the data shown. Was this decrease seen in other experiments, which might suggest a role for Tom1 in the Tollip:Parkin proximity/interaction? Can the authors comment on why biotinylation of wild-type HA-Parkin (and also Parkin mutants) was markedly decreased following AO in 4E, but not 3A?

Response: Yes, we would expect an increase in biotinylation following AO treatment, and this is supported by the new coIP data in Fig 3C, which shows an interaction specifically after AO induced mitochondrial damage. However, multiple repeats of the BioID experiments in Fig 3A are subtly variable and are therefore difficult to quantify. These data are therefore meant to be a qualitative assessment of interactions. In addition, due to the 6 hour labelling period of the

BiOID experiments, it represents an accumulation of signal over time, rather than a snapshot in time for which the coIP indicates.

We partially agree with the reviewer's assessment of the Tollip-Parkin interaction in Tom1 KO cells. However, much of the differences are likely due to the overall expression level of the HA-Parkin in the Tom1 KO cells. As you can see from the lysate lane, levels of HA-Parkin are reduced compared to wild-type and Atg5 KO cells. This may account for the decreased biotinylation in the pulldown lanes. Our overall conclusion from these data is that the Tollip-Parkin interaction is retained in the absence of Tom1. It is difficult to take quantitative data from these experiments, but we know from data in Fig 6 that loss of Tom1 disrupts Tollip translocation to a LAMP1 compartment, therefore there may be an indirect effect on the Tollip-Parkin interaction due to this. In addition, there may be compensating mechanisms coming from expression of the highly related Tom1 family member, Tom1L2, which can replace some of Tom1's function (Tumbarello et al., Nat Cell Biol, 2012).

The experiments associated with Figure EV4B were done in conditions with a 24 hour labelling period, while those in Figure 3A were only labelled for 6 hours. Because of the longer labelling in the presence of AO for EV4B, there is an increase in Parkin degradation due to activation of the mitophagy pathway. Thus, we agree it is difficult to assess the Tollip-Parkin interaction in the AO-treated samples in EV4B. These were challenging experiments due to the instability of the various Parkin mutants, therefore we had to extend the labelling period in order to confidently assess the biotinylation of Parkin.

4. Previous reports have shown that R42P in the Ubl domain fails to target mitochondria (Lee et al JCB 2010).

Response: It is true that a number of studies disagree with one another regarding the ability of the R42P mutant of Parkin, as well as other mutants, to translocate to the mitochondria. However, there have been published studies that indicate the translocation of the R42P mutant to mitochondria in response to damage, although in some cases with a partial reduction compared to wild-type (Narendra, D.P. et al., PLOS Biology, 2010; Fiesel, F.C. et al., Hum Mutat, 2016). This is the primary reason we included these data in now Figure EV4A to illustrate in our hands the localisation pattern of each of the mutants, which shows at least partial translocation of R42P in response to AO treatment. We haven't quantified these data, so cannot make conclusions about the efficiency of this translocation, but we feel this analysis is out of the scope of this publication.

5. The authors conclude that the recruitment of Parkin to Tollip does not require its E3 ligase activity. However, once recruited, Parkin is proposed to ubiquitinate resident proteins in the endosome to mediate further lysosomal trafficking. It would be interesting to understand how Parkin activity is triggered. Do the MDVs contain phospho-S65 ubiquitin for example?

Response: This is a very interesting point as the mechanisms of Parkin activation on MDVs hasn't yet been clearly established. It would be important to determine if this mechanism is similar to the PINK1-Parkin mechanism of mitophagy induction. Based on the reviewer's comments, we have attempted to address this by determining whether pS65 ubiquitin decorates Tom20 MDVs in SH-SY5Y cells. This has proven to be challenging due to antibody limitations and sensitivity. While we can clearly assess pS65 ubiquitin on whole mitochondria following damage, we cannot thus far be confident individual MDVs are decorated with this ubiquitin species (see figure below). This will require further investigation, which we believe, while interesting, is out of the scope of this study.

Specific

1. Hela should be HeLa throughout
2. Page 6: "Further implementing..." "supporting"?
3. Western blot to confirm expression of R78A mutant alongside wt or the CUD domain mutant should be shown in Figure S4D.
4. Page 13, "Surprisingly, Parkin depletion lead..." should read "led".
5. Molecular weight markers should be added to their Western images.

Response: All of the above points have been fixed. For point 3, these data are now in new Appendix Figure S2D. Thank you for bringing these to our attention.

Thank you for submitting a revised version of your manuscript. It has now been seen by the original referees, whose comments are shown below.

As you will see they find that all criticisms have been sufficiently addressed and recommend the manuscript for publication. Please note that referee #2 suggests some minor text modifications.

In addition, there are a few editorial issues concerning the text and the figures that I need you to address before we can officially accept the manuscript.

REFEREE REPORTS

Referee #1:

The authors have responded to most of my comments and concerns, and the images are clearer, and the study is more complete. I think their main point about Tollip as a regulator of Tom20 MDVs is sound, and the links to Parkin are important, and in some ways, unexpected. They removed the most concerning figure describing the turnover of Tom20, which is probably a good thing, although it then leaves open the function of this pathway (which is an ongoing issue in that field). There is obviously much left to do to understand these mechanisms, and I still find parts of their model confusing, but they have answered my concerns. I think this will be a story of interest to the readership of EMBO, particularly those following the field of mitochondrial dynamics, Parkinsons research, and immunology (although not specified here, there are many who follow Tollip in the Toll pathway).

Referee #2:

The authors have addressed my remaining concerns sufficiently. The authors have included new data on mitochondrial functionality, albeit I was hoping for more sensitive assays in this regard. The determination of ATP levels is quite insensitive and would only reveal major disturbances of mitochondrial dysfunction. As this is not the main message of the study and as the authors have argued well in this matter, I am fine with this additional experiments. Future studies may address this in more detail. I feel that this nice piece of work should be published as it is of major general interest. The role of Tollip and PARKIN for MDV trafficking to lysosomes via the endosomal pathway is of broad interest also to other fields.

Referee #3:

The authors have addressed each of my points and provided new data to support their conclusions.

My only suggestion would be to reword the sentence, "To confirm if Tollip is in tight complex with Parkin....." to "To test if Tollip associates with Parkin in cells, we performed BioID proximity-dependent labelling." This would avoid over-interpretation of the BioID analyses, which informs proximity, and not necessarily a "tight" interaction. Indeed, as recognised by the authors, the interaction between Parkin and Tollip is likely transient (supported by their inefficient co-precipitation even when over-expressed).

Please see below our point-by-point response to the reviewer's comments:

Referee #1:

The authors have responded to most of my comments and concerns, and the images are clearer, and the study is more complete. I think their main point about Tollip as a regulator of Tom20 MDVs is sound, and the links to Parkin are important, and in some ways, unexpected. They removed the most concerning figure describing the turnover of Tom20, which is probably a good thing, although it then leaves open the function of this pathway (which is an ongoing issue in that field). There is obviously much left to do to understand these mechanisms, and I still find parts of their model confusing, but they have answered my concerns. I think this will be a story of interest to the readership of EMBO, particularly those following the field of mitochondrial dynamics, Parkinsons research, and immunology (although not specified here, there are many who follow Tollip in the Toll pathway).

Response: We would like to thank the reviewer for their positive comments on the additional data that has been included, their thorough assessment of the manuscript throughout the review process, and their appreciation for the impact of this study.

Referee #2:

The authors have addressed my remaining concerns sufficiently. The authors have included new data on mitochondrial functionality, albeit I was hoping for more sensitive assays in this regard. The determination of ATP levels is quite insensitive and would only reveal major disturbances of mitochondrial dysfunction. As this is not the main message of the study and as the authors have argued well in this matter, I am fine with this additional experiments. Future studies may address this in more detail. I feel that this nice piece of work should be published as it is of major general interest. The role of Tollip and PARKIN for MDV trafficking to lysosomes via the endosomal pathway is of broad interest also to other fields.

Response: We very much thank the reviewer for their positive comments on the additional data

that has been included and the overall impact of the study. We also thank the reviewer for their thorough assessment of the manuscript throughout the review process.

Referee #3:

The authors have addressed each of my points and provided new data to support their conclusions.

My only suggestion would be to reword the sentence, "To confirm if Tollip is in tight complex with Parkin....." to "To test if Tollip associates with Parkin in cells, we performed BioID proximity-dependent labelling." This would avoid over-interpretation of the BioID analyses, which informs proximity, and not necessarily a "tight" interaction. Indeed, as recognised by the authors, the interaction between Parkin and Tollip is likely transient (supported by their inefficient co-precipitation even when over-expressed).

Response: We thank the reviewer for their positive comments on the additional data that has been included and their thorough assessment of the manuscript throughout the review process. We appreciate the reviewers concerns with the phrase 'tight complex' on page 7 of the manuscript and agree this is an overstatement of what our data indicates. Therefore, we have adjusted the wording as recommended by the reviewer, with inclusion of a minor additional change to the sentence to ensure good flow.

Accepted

10th March 2020

I am pleased to inform you that your manuscript has been accepted for publication in the EMBO Journal.

Corresponding Author Name: David A Tumbarello

Journal Submitted to: The EMBO Journal

Manuscript Number: EMBOJ-2019-102539